# An Ensemble Model based on Deep Learning and Data Preprocessing for Short-Term Electrical Load Forecasting

Yamin Shen [1], Yuxuan Ma [1], Simin Deng [1], Chiou-Jye Huang [2,*] and Ping-Huan Kuo [3,4,*]

1 School of Electrical Engineering and Automation, Jiangxi University of Science and Technology, Ganzhou 341000, China; 6720180578@mail.jxust.edu.cn (Y.S.); 6120190479@mail.jxust.edu.cn (Y.M.); 6720200723@mail.jxust.edu.cn (S.D.)
2 Department of Data Science and Big Data Analytics, Providence University, Taichung 43301, Taiwan
3 Department of Mechanical Engineering, National Chung Cheng University, Chiayi 62102, Taiwan
4 Advanced Institute of Manufacturing with High-tech Innovations (AIM-HI), National Chung Cheng University, Chiayi 62102, Taiwan
* Correspondence: cjh1007@pu.edu.tw (C.-J.H.); phkuo@ccu.edu.tw (P.-H.K.); Tel.: +886-4-2632-8001 (C.-J.H.); +886-5-2720411 (P.-H.K.)

**Abstract:** Electricity load forecasting is one of the hot concerns of the current electricity market, and many forecasting models are proposed to satisfy the market participants' needs. Most of the models have the shortcomings of large computation or low precision. To address this problem, a novel deep learning and data processing ensemble model called SELNet is proposed. We performed an experiment with this model; the experiment consisted of two parts: data processing and load forecasting. In the data processing part, the autocorrelation function (ACF) was used to analyze the raw data on the electricity load and determine the data to be input into the model. The variational mode decomposition (VMD) algorithm was used to decompose the electricity load raw-data into a set of relatively stable modes named intrinsic mode functions (IMFs). According to the time distribution and time lag determined using the ACF, the input of the model was reshaped into a $24 \times 7 \times 8$ matrix $M$, where 24, 7, and 8 represent 24 h, 7 days, and 8 IMFs, respectively. In the load forecasting part, a two-dimensional convolutional neural network (2D-CNN) was used to extract features from the matrix $M$. The improved reshaped layer was used to reshape the extracted features according to the time order. A temporal convolutional network was then employed to learn the reshaped time-series features and combined with the fully connected layer to complete the prediction. Finally, the performance of the model was verified in the Eastern Electricity Market of Texas. To demonstrate the effectiveness of the proposed model data processing and load forecasting, we compared it with the gated recurrent unit (GRU), TCN, VMD-TCN, and VMD-CNN models. The TCN exhibited better performance than the GRU in load forecasting. The mean absolute percentage error (MAPE) of the TCN, which was over 5%, was less than that of the GRU. Following the addition of VMD to the TCN, the basic performance of the model was 2–3%. A comparison between the SELNet model and the VMD-TCN model indicated that the application of a 2D-CNN improves the forecast performance, with only a few samples having an MAPE of over 4%. The model's prediction effect in each season is discussed, and it was found that the proposed model can achieve high-precision prediction in each season.

**Keywords:** electricity load forecasting; deep learning; variational mode decomposition; two-dimensional convolutional neural network; temporal convolutional networks



## 1. Introduction

As the smart grid's development foundation, the demand for electricity load is one of the hot concerns in the current electricity market. There are many methods to forecast the electricity load. The prediction can be realized by extracting time features from historical load data and extracting relevant features from influencing factors such as economy,

society, and weather. [1] An accurate prediction model can provide a reliable plan for the power operation system and maintain its economical operation. New energy and renewable energy power generation methods have not been put into use on a large scale. [2] Conventional power generation methods will consume many fossil fuels and other energy resources and emit polluting gases to destroy the atmosphere. In the development of power systems, improving load forecasting accuracy has always been an important task. If the load forecasting results are lower than the truth, the system's planned installed capacity will be insufficient, and it will not meet the social power demand. If the load forecasting results are higher than truth, it will lead to the operating efficiency of power generation or transmission equipment decline, causing energy waste. Therefore, if it is possible to accurately forecast the demand for short-term power load in the future, it will save unnecessary energy losses [3] and protect the atmosphere from pollution as much as possible. With the development of distributed generation and grid-connected, the power load forecasting models have become an imperative requirement.

According to the forecasting horizon, load forecasts can be divided into long-term load forecasts [4] of one to twenty years in advance, medium-term load forecasting [5] of one week to one year in advance, short-term load forecasts [6] of one hour to one week in advance, and a few minutes to one hour in advance. [7] The demand for forecasting power load one day in advance is strongest in the current electricity market. [2] In the early stage of power load forecasting, simpler models such as regressive models were used for time-series forecasting. For example, Moghaddas-Tafreshi et al. [8] used the Linear Regression method, Zheng et al. [9] used the Kalman filtering, and Souza [10] used Auto-Regressive Integrated Moving Average. However, these models are effective when dealing with linear forecasting problems, but they are not effective when dealing with complex nonlinear time series such as load data.

The use of machine learning for time-series forecasting is becoming increasingly popular. Bhatia et al. [11] proposed a hybrid adaptive boosting and eXtreme gradient boosting (XGBoost) model and used the rolling forecast method for load forecasting. They verified the effect of the proposed model on the German electricity market. Yang et al. [12] selected optimal input features and combined the autocorrelation function (ACF) and the least-squares support vector machine (LSSVM) to establish a hybrid model called AS-GCLSSVM for power load forecasting. They used the ACF to select information input variables, LSSVM for prediction of electrical load, and the grey wolf optimization algorithm and cross validation for optimizing the parameters in the LSSVM. The model in that experiment was used to predict the power load for the next week and in half hour, and the authors' proposed power load data set was used in three regions; their model was compared with nine other models. They found that the AS-GCLSSVM significantly increased the accuracy of short-term power load forecasting. However, this model also had some shortcomings, such as a high time consumption and complex algorithms. Ahmad et al. [13] proposed a short-term power load forecasting model based on machine learning and a meta-heuristic algorithm. They used the XGBoost model and a decision tree for feature selection; they applied their model to a real-time power data set (between January 2017 and December 2019) from New England's independent power system and found that the improved extreme learning machine model optimized with a genetic algorithm (ELM-GA) achieved a classification accuracy of approximately 96.42%. Because the classification probability cannot intuitively represent the correlation between different moments in a continuous time series, Toubeau et al. [14] sampled the predicted multivariate distribution based on a correlation strategy, which improved the dependence of the time series in classification probability prediction.

With the development of artificial neural networks, Zahedi et al. [15] used an adaptive fuzzy neural network to model the electricity demand in Ontario province, Canada, and used statistical methods such as Pearson correlation to filter the input. Zahedi thinks that the shortcoming is that it usually takes a lot of time to build models using fuzzy neural networks. For some nonlinear systems, the fuzzy neural network also needs to be combined

with Fourier and other regression models [16]. Park [17] proposed a similar day selection model based on reinforcement learning and a Back Propagation Neural Network (BPNN) load prediction model based on similar days to improve load forecasting accuracy. The results show that the proposed similar day selection model has an accuracy of 97.19% to determine a similar load date. For long-term load forecasting, Nyandwi et al. [18] proposed a deep feedback neural network model based on temperature regulation. They made a year-long forecast of the New York Independent System Operator. For short-term load forecasting, Khan et al. [19] used a feature filtering method based on hybrid random forest and recursive feature elimination methods to select network inputs and combined this hybrid method with a deep neural network to forecast loads for a week. The accuracy of the model after the input features had been filtered was considerably higher than that of a traditional CNN model.

Atef et al. [1] used deep-layered unidirectional long short-term memory (LSTM) and bidirectional LSTM networks to predict the impact of power load consumption. They compared LSTMs with different depths using two stacking methods and used optimization algorithms to tune each model. The experimental results indicated that the deep stacking LSTM layer consumed almost twice as much time as the single-layer model. Kumar et al. [20] proposed a model based on LSTM and a gated recurrent unit (GRU) to solve the problem of nonlinearity and seasonality in power load data forecasting.

On the other hand, some researchers believe that the power load is an unstable time series, so it is necessary to use the preprocessing method to stationary processing the power load data. Sun et al. [21] first used empirical mode decomposition (EMD) to stationary processing the power load data, combined with sample entropy to filter the input features, according to phase space reconstruction get input and output. This method of screening features is to rely on the sample entropy as a loss function to determine whether the features meet the experimental conditions, which can improve the prediction accuracy to a certain extent. Wu et al. [22] combined VMD and FA-KELM to improve the accuracy of forecasting photovoltaic power generation in the day and proposed a new hybrid model of photovoltaic power generation output prediction interval, using VMD to decompose the photovoltaic power sequence, and each decomposed component obtain a prediction result through FA-KELM, and continue to sum the prediction results, through cubic spline interpolation can obtain the confidence interval of predicted photovoltaic power. Experiments show that the mixed-use of VMD and FA-KELM can effectively construct the best confidence interval and obtain a more accurate photovoltaic power output prediction interval.

The main contributions of this research are as follows:

1. In the experiment, the ACF was used to analyze the raw data on the electricity load. Raw data on the hourly electricity load of the previous week were selected as inputs for the model. The selected load data were then reshaped into a $24 \times 7$ matrix, with each column representing the electricity load for the 24 h of a day and each row representing the electricity load at the same time in the previous week.
2. The VMD algorithm is used to decompose the load data into 8 IMFs with different center frequencies; each mode is expanded into a $24 \times 7$ matrix according to ACF and concatenated. The input data $M$ become $24 \times 7 \times 8$. The advantage is that the various modes are more stable than the original power load. Still, the influence of seasonal factors on the model's accuracy can be reduced, making the model universal in seasons. It can also make it easier for 2D-CNN to extract temporal features.
3. A 2D-CNN was used to extract the features from the hourly load change and the load change in the same period of the week on expanded matrices.
4. During the experiment, an improved Reshape network layer was used for dimensional transformation. The three-dimensional tensor of the output of the last layer of the 2D-CNN network was connected by slices to keep the channel content unchanged, and each slice was processed in time series. Reduce the dimensionality and obtain a new set of two-dimensional tensors with time meaning. Then input this group of

tensors into the TCN network for time series learning combined with full connection to achieve the final prediction.

5.  This study combines VMD, CNN, and TCN to forecast next-day hourly electricity load data. To verify the effectiveness of SELNet, we compared it with the GRU, TCN, VMD-TCN, and VMD-CNN models; the results are discussed in Section 3.

## 2. Materials and Methods

### 2.1. Variational Mode Decomposition

Unlike classic EMD [23], VMD uses a nonrecursive form to complete signal decomposition, wherein the decomposed subsignals are extracted simultaneously. VMD relaxes the bandwidth limitations and reconstruction fidelity constraints during signal decomposition, and therefore, it is more robust in terms of noise sensitivity. VMD determines the mode and the corresponding center frequency set according to the constraint conditions to reconstruct the original signal.

To determine the bandwidth limitation, each mode must obtain the corresponding analytical signal through the Hilbert transform, after which the analytical signal is mixed with the exponential form of the respective estimated center frequency to transfer the current frequency to the baseband. Finally, expression of the bandwidth limitation is completed through Gaussian smoothness, as shown in (1).

A constraint function is introduced into VMD [24,25], which uses the square norm of the gradient to achieve Gaussian smooth estimation, as shown in the following Equation (1):

$$\min_{\{u_k\},\{\omega_k\}} \left\{ \sum_k \left\| \partial t \left[ \left( \delta(t) + \frac{j}{\pi t} \right) * u_k(t) \right] e^{-j\omega_k t} \right\|_2^2 \right\} \\ s.t. \sum_k u_k = f \tag{1}$$

where $\{u_k\}, \{\omega_k\}$ represent all the mode sets and their center frequencies, respectively.

The augmented Lagrangian and square norm are used to reduce the influence of noise; the augmented Lagrangian function is given by the following equation:

$$L(\{u_k\}, \{\omega_k\}, \lambda) := \alpha \sum_k \left\| \partial t \left[ \left( \delta(t) + \frac{j}{\pi t} \right) * u_k(t) \right] e^{-j\omega_k t} \right\|_2^2 \\ + \left\| f(t) - \sum_k u_k(t) \right\|_2^2 + \left\langle \lambda(t), f(t) - \sum_k u_k(t) \right\rangle \tag{2}$$

The alternate direction method of multipliers [26] is used to optimize the solution of the Lagrangian saddle point to obtain the alternate update equations of $u_k$ and $\omega_k$.

Convert the update mode $u_k$ into a problem of minimizing the equation, as in Equation (3):

$$u_k^{n+1} = \underset{u_k \in X}{argmin} \left\{ \alpha \left\| \partial t \left[ \left( \delta(t) + \frac{j}{\pi t} \right) * u_k(t) \right] e^{-j^2 \omega_k t} \right\|_2^2 + \left\| f(t) - \sum_i u_i(t) + \frac{\lambda(t)}{2} \right\|_2^2 \right\} \tag{3}$$

Convert the update $\omega_k$ into a problem of minimizing the equation, as in Equation (4):

$$\omega_k^{n+1} = \underset{\omega_k}{argmin} \left\{ \left\| \partial t \left[ \left( \delta(t) + \frac{j}{\pi t} \right) * u_k(t) \right] e^{-j^2 \omega_k t} \right\|_2^2 \right\} \tag{4}$$

The modes decomposed by VMD can balance the errors, and each mode corresponds to an orthogonal basis, so the modes are largely disjoint.

### 2.2. Two-Dimensional Convolutional Neural Network

The two-dimensional convolutional neural network (2D-CNN) [27,28] is defined because its convolution kernel has two moving calculation directions, compared with a one-dimensional convolutional neural network (1D-CNN), as show in Figure 1. It can add a feature map of one dimension and extract feature information of two dimensions at the same time. In this paper, 2D-CNN is used for convenience and is represented by CNN [29,30]. Convolutional neural networks include convolutional layers and pooling layers. The convolutional layer is used to extract local features, and the pooling layer compresses the extracted local features and reduces the number of learning parameters. The equation of the 2D-CNN convolution process is given in (5):

$$y_{kl} = \sum_{i=2k-1}^{2k} \sum_{j=2l-1}^{2l} x_{ij} \omega_{mn} \tag{5}$$

where $y_{kl}$ represents the output of the filter, $x_{ij}$ represents the input, $w_{mn}$ represents the weight of the filter, and $k \in \{1,2\}, l \in \{1,2\}, m = i - 2 \times (k-1), n = j - 2 \times (l-1)$.

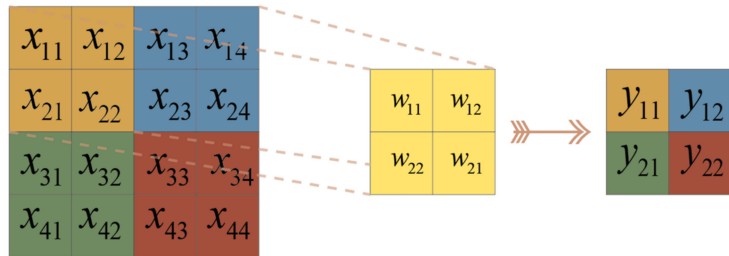

**Figure 1.** Traditional two-dimensional convolutional neural network (2D-CNN) convolution process.

CNNs have been widely used in time series and image processing. In this experiment, the CNN model was used to extract features from the expanded matrices, and in combination with a TCN, it was used for electricity load prediction.

### 2.3. Temporal Convolutional Network

Many feasible models have emerged in the field of time series forecasting, such as RNN [31], LSTM [32], and GRU [20]. However, in recent years, the appearance of TCN as an improved architecture for CNN has successfully defeated models such as RNN in time series processing in many fields and has shown excellent performance. TCN combines the characteristics of causal convolution, dilated convolution, and residual block [33], so TCN has the following advantages:

Since TCN uses causal convolution, there is a causal relationship between convolutional network layers and layers. It is a one-way structure with strict time constraints.

The effective window of the dilated convolution will increase exponentially as the sampling rate of the dilated convolution increases. Therefore, the TCN can use fewer layers to obtain a large receptive field [34].

Residual linking has been proven to be an effective method for training deep networks. It allows the network to transmit information in a cross-layer manner, reduces problems such as gradient disappearance, and enhances the model's robustness.

#### 2.3.1. Causal Convolution

In dealing with time series, it is necessary to predict future data through historical data, so it has strong causal constraints [35]. Causal convolution satisfies this characteristic. Figure 2 shows a causal example of causal convolution:

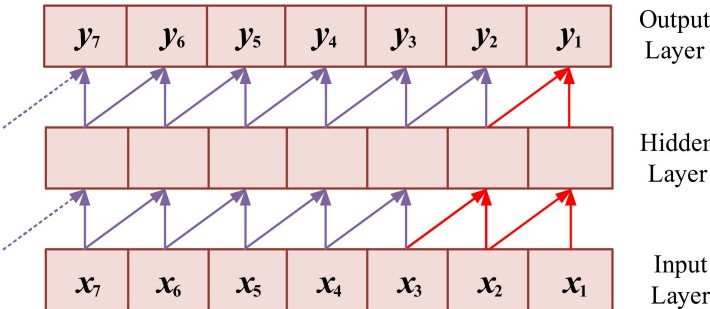

**Figure 2.** Causal convolution.

It is not difficult to find from Figure 2 that to get $y_1$ corresponding to time 1, only the input features at $x_1, x_2, x_3$ are allowed to obtain. In this figure, the convolution kernel is a $2 \times 1$ convolution kernel. It moves in the first layer one by one, and the padding is done on the leftmost layer to get the second layer, and then pass it down by layer to get the causal effect. However, causal convolution faces a problem. If you want to increase the receptive field, you need to increase the convolution kernel's size or deepen the network depth. Increasing the size of the convolution kernel will increase the model parameters and increase the training time. Deepening the network depth will bring about problems such as the disappearance of gradients, unstable training, and convergence difficulty [36]. Therefore, combining causal convolution with dilated convolution is a good choice.

### 2.3.2. Dilated Convolution

The dilated convolution role is to replace the pooling layer in the traditional convolutional neural network [37], which can avoid the increase in the calculation and increase the receptive field.

Figure 3 shows the addition of dilated convolution solves the problems faced by causal convolution. In the figure, the dilated rate is 1 and 2. The size of the convolution kernel is 2. The introduction of dilated convolution enables causal convolution to increase the receptive field without increasing the network's depth and the number of parameters that need to be learned. In this way, the characteristics between inputs in a larger time range can be learned. The difference between dilated convolution and causal convolution is the addition of dilatation coefficients 1 and 2 to dilated convolution, which expands the receptive field of $y_1$ to $x_4$ and reduces the amount of calculation.

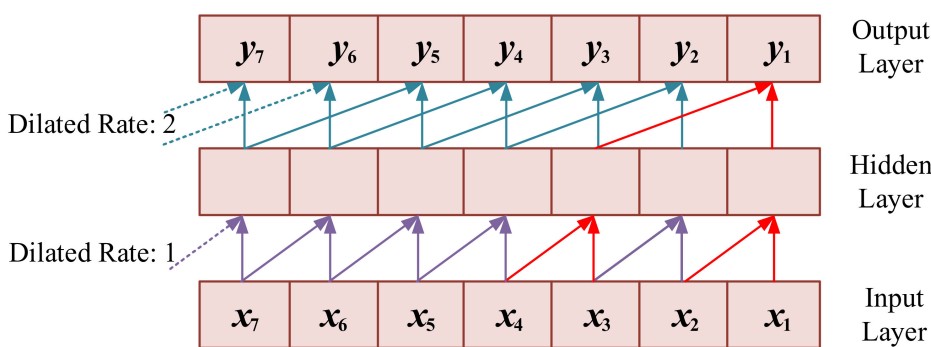

**Figure 3.** Causal Convolution with dilations.

### 2.3.3. Residual Block

The residual block concept is derived from the model used by the Microsoft team in the ImageNet image vision competition in 2015. ResNet was introduced in detail in a paper published by this team [38]. As the number of deep learning networks became deeper and deeper, deeper neural networks would be oversaturated and degraded. Therefore, the results of shallow network training may be better than those of deep network training. To

solve this problem, the shallow network is identically mapped to the deep network and the residual characteristics are constantly updated to ensure that the training effect of the deep network is not worse than that of the shallow network.

ResNet is based on an identity mapping function, $H(x) = x$.

$$y = F(x, \{W_i\}) + x \qquad (6)$$

where x and y in Equation (6) represent the input and output matrices, and the function $F(x, \{W_i\})$ represents the residual feature map that needs to be learned. The structure of the specific residual block is shown in Figure 4.

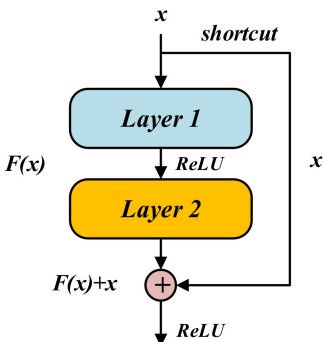

**Figure 4.** Residual block.

This kind of residual learning also uses a shortcut to connect the input and output to achieve identity mapping. The training parameters will not increase, and the shortcut connect can be used to solve problems such as gradient disappearance.

2.3.4. Proposed Model

A model architecture based on hybrid VMD, CNN, and TCN proposed in this paper is shown in Figure 5. Using signal decomposition method to decompose the original time-series signal ensures that the seasonal variation trend in power load has negligible influence on the prediction accuracy of the model; this considerably reduces the complexity of the model. As a new signal decomposition method, VMD performs better than classic EMD. The model uses the classic and effective CNN to extract features between different days and the same time period to increase the prediction accuracy. Compared with other models, TCN exhibits better performance in time-series forecasting. The TCN model can not only contain more historical features but also reduce the amount of calculation considerably. The model contains three convolutional layers, a Reshape layer, and a TCN layer. Among them, the size of the convolution kernel of each convolution layer is (12,5), (12,6), (7,7), and the number of filters is 72, 84, 96. The function of the Reshape layer is to reshape the output data ($24 \times 7 \times 96$) of the last layer of CNN into ($168 \times 96$) according to the data of one week and then input it into the TCN network for training. According to this structure, TCN can use continuous-time sequence learning time features, the number of filters used in the TCN network is 60, and the hidden layer has two layers. The expansion coefficients of the hidden layer are 1 and 2. The output result is fully connected to the Dense layer through Flatten. Finally, the entire day's prediction results are output through the Dense layer. The specific parameters in the network layer can be referred to in Table 1 below.

Figure 6 shows the flow chart of the experiment. Considering that the electricity load is a non-stationary time series, VMD is used to decompose the original electricity load into 8 relatively stable IMFs. According to the ACF, the hourly electricity load data of the forecast day is dependent on not only the characteristics of the hourly load change but also the same-time load change during the week. The input is reshaped into a $24 \times 7 \times 8$ matrix $M$, where 24, 7, and 8 represent 24 h, 7 days, and 8 IMFs, respectively. The CNN is used to extract the features from $M$, and the extracted features continue to be reshaped and reduced into a time series, after which the TCN is used for time-series processing for prediction. The

addition of VMD, CNN, and TCN enhances the performance of SELNet; this is discussed in Section 3.

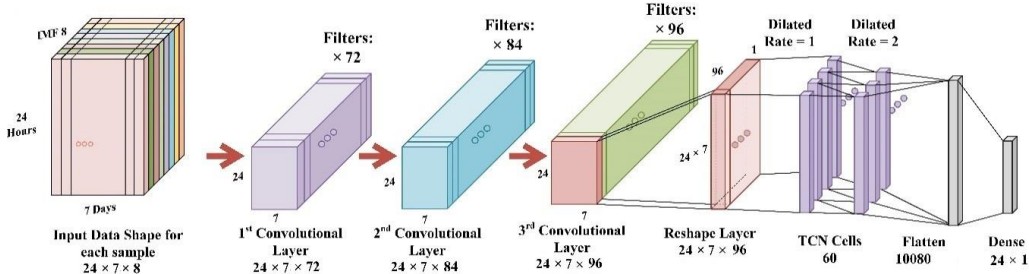

**Figure 5.** The architecture of our proposed model.

**Table 1.** The major parameter list for each layer.

| Layer | Filters | Dilation | Output Shape |
|---|---|---|---|
| Input | - | - | $(24 \times 7 \times 8)$ |
| 1st CNN | 72 | - | $(24 \times 7 \times 72)$ |
| 2nd CNN | 84 | - | $(24 \times 7 \times 84)$ |
| 3rd CNN | 96 | - | $(24 \times 7 \times 96)$ |
| Reshape | - | - | $(168 \times 96)$ |
| TCN | 60 | [1,2] | $(168 \times 60)$ |
| Flatten | - | - | $(10,080 \times 1)$ |
| Dense | - | - | $(24 \times 1)$ |

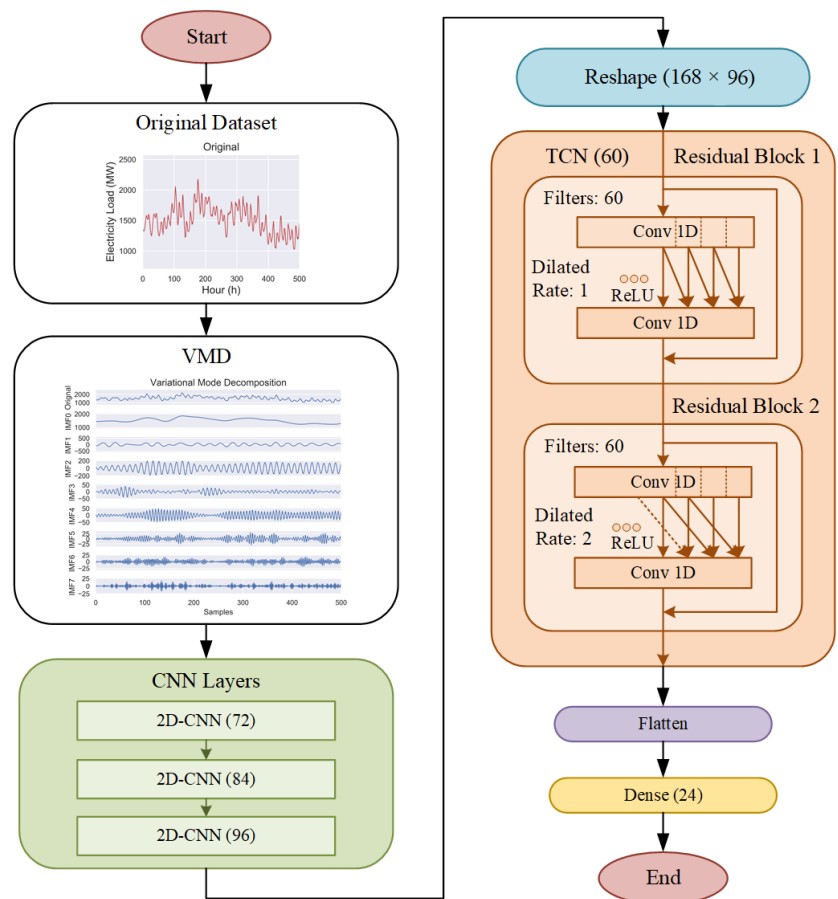

**Figure 6.** Flow chart of our proposed model.

## 3. Experimental Results

This experiment selected the eastern region of the American Electric Reliability Council of Texas (ERCOT) power grid as the research object. As shown in Figure 7, the ERCOT power network covers about 75% of Texas, and the ERCOT power market manages more than 90% of the users in Texas. Because Houston is the main resource node in the eastern region, energy demand accounts for the vast majority of the eastern region. This experiment selects ERCOT's eastern region's power report [39] as the experimental database. This experiment's data use the hourly data of 1057 days from January 1, 2015 to November 30, 2017 as a sample. The experiment randomly shuffles the input and output into samples, improving the generalization of the model. So, 80% of the total sample was selected as the training set and 20% as the test set. Considering that the geographic range of Texas in the Northern Hemisphere is approximately 25°50′–36°30′ north latitude, 93°31′–106°38′ west longitude, according to the local climate change and local people's perception, we use the local Texas climate to divide the seasons. The approximate seasons are divided into the following Table 2.

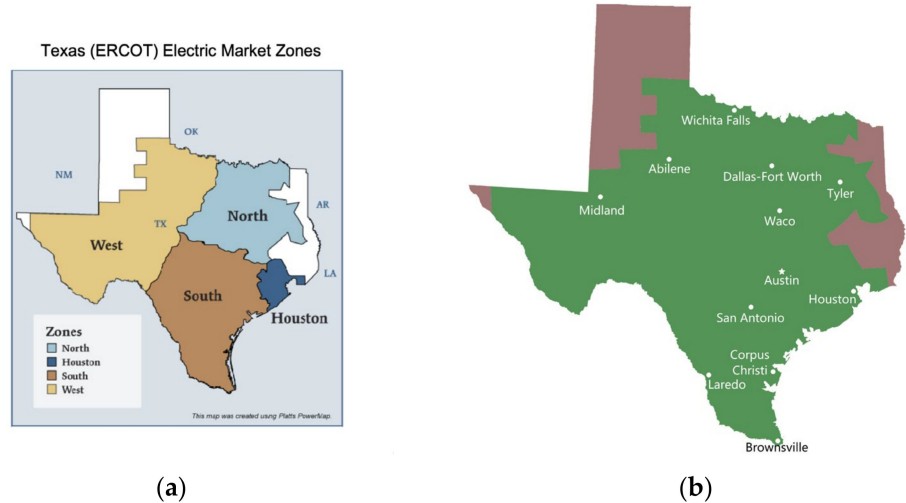

(**a**)                    (**b**)

**Figure 7.** (**a**). ERCOT Load Zone Map [40]; (**b**). The ERCOT grid covers approximately 75% of the land area in Texas [41].

**Table 2.** Division of time range of four seasons in the state of Texas.

| Season | Date Range |
|---|---|
| Spring | 3.21–6.21 |
| Summer | 6.22–9.22 |
| Autumn | 9.23–12.21 |
| Winter | 12.22–3.20 |

*Data Preprocessing*

In this experiment, the VMD algorithm is used to decompose the original load data into eight relatively stable IMFs. The number of decomposed modes and the optimal center frequency are obtained through a grid search algorithm. It is verified that the loss value of each mode after decomposition is within the allowable range, among which, $K = 8$, $\alpha = 505$ the decomposition effect of the original electricity price is shown in Figure 8, where the decomposition effect is shown for the entire year of 2015. It can be seen from Figure 8 that the low-frequency signal has a smoother trend than the original signal, and the trend of the low-frequency part is roughly in line with the original signal, indicating that the VMD decomposition can effectively separate noise and is more convenient for prediction. From the figure, the low-frequency part of the four seasons of a complete year fluctuates more smoothly, which is one reason why the experiment chose to forecast regardless of the season.

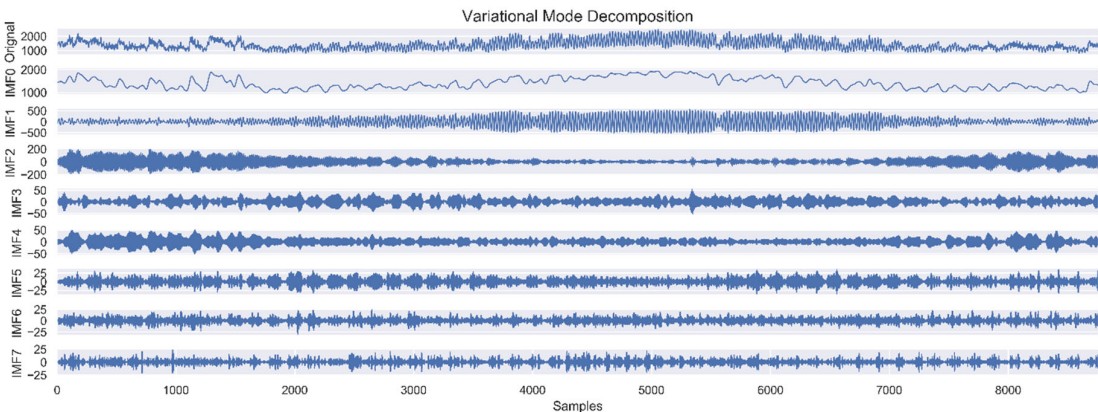

**Figure 8.** Variational Mode Decomposition.

In the feature selection part, because the experiment mainly uses load data for training, in the time series analysis process, the autocorrelation function (ACF) is often used to evaluate the degree of influence between the events before and after the event so that this experiment will be in the database. The hourly lag of 5 weeks and the daily lag of 2 weeks selected for the same time, as evaluated using the ACF, are shown in Figure 9.

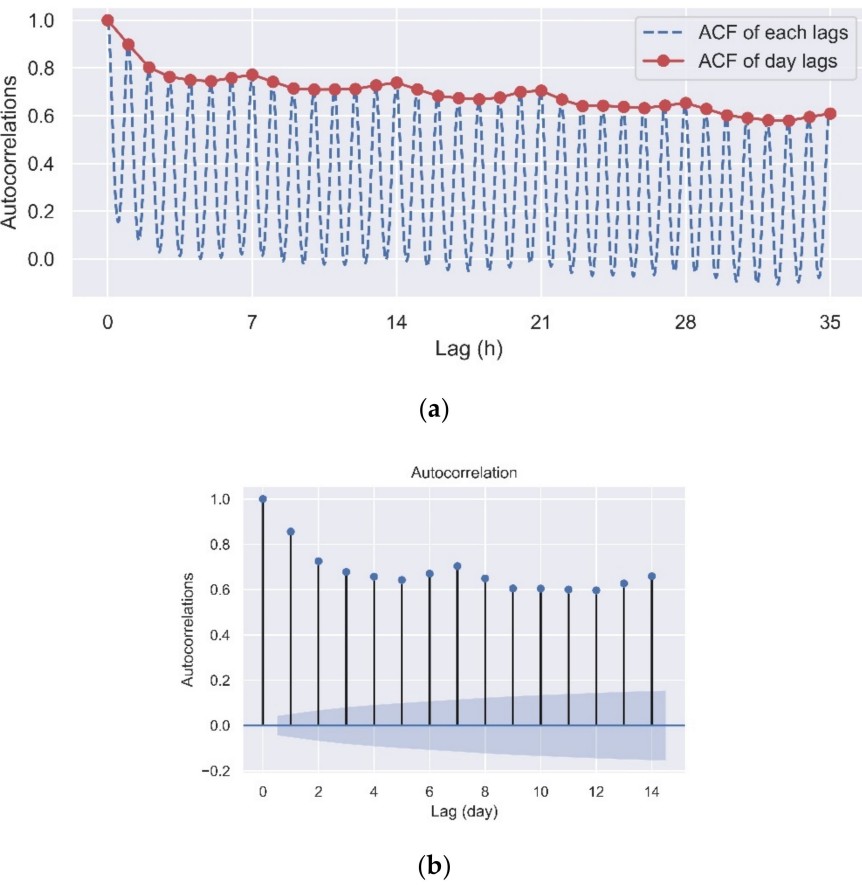

**Figure 9.** Feature selection. (**a**) Autocorrelation function applied in a 35-consecutive-hours. (**b**) Autocorrelation function is applied in the same hour in two weeks (12:00 a.m. as an example).

It can be seen from the blue line in Figure 9a that the autocorrelation coefficient reaches its peak when Lag is one day, which indirectly reflects the high correlation between load changes at the same time, so the input data are expanded to $7 \times 24$ matrices so that the

model can not only extract the relevant features of the change between hourly load in a day but also extract the change characteristics of the same hour in a week. When Lag reaches the first week, the second week, etc., it can be found that the power load has a certain weekly change trend, as shown by the red line in Figure 9a. Therefore, the power load data of the previous seven days are used as input data, according to which is the most suitable. Figure 9b selects the autocorrelation coefficient image made by the time series of the power load composition at the same time at 12:00 a.m. in the middle time, and displays it in the two states of Lag, verifying the conclusion from Figure 9a at the same time the load changes are highly correlated, with weekly trends, etc.

To evaluate the accuracy of the model, the commonly used Mean Absolute Percentage Error (MAPE) and Root Mean Square Error (RMSE) in the field of time series forecasting are used to evaluate the accuracy of the model. The equation is as in the following equations:

$$MAPE = \frac{100}{N} \sum_{i=1}^{N} \left| \frac{y_i - \hat{y}_i}{y_i} \right| \tag{7}$$

$$RMSE = \sqrt{\frac{\sum_{i=1}^{N} (y_i - \hat{y}_i)^2}{N}} \tag{8}$$

where $y_i$ represents the true value, $\hat{y}_i$ represents the predicted value, and $N$ represents the number of samples.

This experiment is for daily power load forecasting. Therefore, MAPE and RMSE can be obtained by comparing the predicted value with real value every day. In order to better show the performance of the proposed model, the MAPE and RMSE of each day in the test set are drawn into quartile Figure, as shown in Figure 10. It can be found from the figure that the proposed model not only has the smallest MAPE and RMSE compared to each model but also the variation range of MAPE and RMSE is relatively concentrated. Table 3 shows the minimum MAPE, maximum MAPE, average MAPE of each model. Table 4 shows the minimum RMSE, maximum RMSE, and average RMSE values of each model. The average MAPE and RMSE are better than the comparison model.

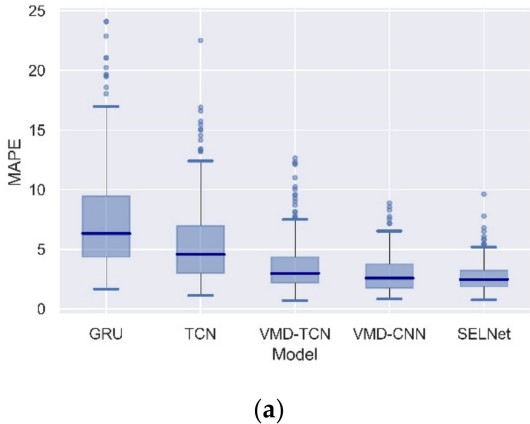
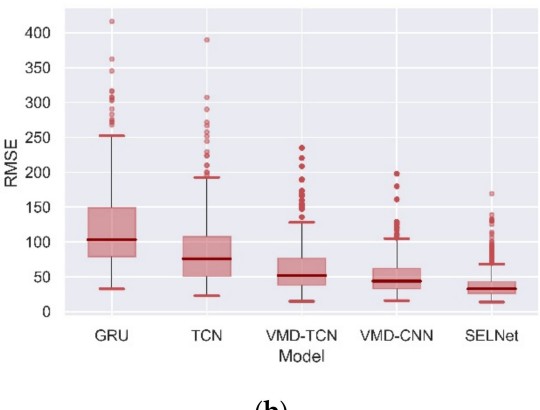

(**a**)　　　　　　　　　　　　　　　　(**b**)

**Figure 10.** (**a**) Quartile map of MAPE for each model; (**b**) Quartile map of RMSE for each model.

**Table 3.** The minimum MAPE, maximum MAPE, average MAPE of each model.

| Model | GRU | TCN | VMD-TCN | VMD-CNN | Proposed |
|---|---|---|---|---|---|
| Min MAPE (%) | 1.665 | 1.113 | 0.690 | 0.830 | 0.746 |
| Max MAPE (%) | 24.092 | 22.508 | 12.645 | 8.880 | 9.624 |
| Average MAPE (%) | 7.621 | 5.544 | 3.680 | 2.986 | 2.738 |

**Table 4.** The minimum RMSE, maximum RMSE, average RMSE of each model.

| Model | GRU | TCN | VMD-TCN | VMD-CNN | Proposed |
|---|---|---|---|---|---|
| Min RMSE | 32.570 | 22.671 | 14.693 | 15.822 | 13.918 |
| Max RMSE | 416.397 | 389.816 | 234.907 | 197.650 | 169.197 |
| Average RMSE | 125.297 | 91.662 | 64.307 | 52.544 | 48.973 |

In Table 5, the prediction results of each model test set MAPE in the range of 0–1%, 1–2%, 2–3%, 3–4%, 4–5%, and days greater than 5% are performed According to statistics, it can be found that 67% of the GRU model has a MAPE value greater than 5%, so the model prediction performance is poor. The TCN model has a MAPE value greater than 5%, accounting for 45%, which is better than GRU, and by combining TCN and VMD algorithms, it can be found that the value of MAPE greater than 5% drops to 20%, which verifies the effectiveness of the VMD algorithm. After the VMD algorithm is decomposed, the data are combined with CNN to extract features from expanded matrices and learn the features through the fully connected layer. The VMD-CNN model has a MAPE greater than 5% and only accounts for 13% of the model performance has been further improved. However, compared with the proposed model whose MAPE is greater than 5%, 6% is still slightly inferior. Moreover, each sample of the proposed model's MAPE is mainly concentrated in 1–3%, which shows that the model can generally predict the power load of the next day well. Therefore, using a CNN to extract features from the expanded matrices, reshaping them into a time series, and applying a TCN to learn the time series and realize the final forecasting is an effective approach.

**Table 5.** MAPE samples statistics in different ranges for each model.

| Model Range (%) | GRU | TCN | VMD-TCN | VMD-CNN | Proposed |
|---|---|---|---|---|---|
| 0–1 | 0 | 0 | 4 | 5 | 2 |
| 1–2 | 2 | 17 | 41 | 70 | 67 |
| 2–3 | 11 | 36 | 65 | 55 | 73 |
| 3–4 | 30 | 39 | 42 | 37 | 46 |
| 4–5 | 26 | 25 | 18 | 17 | 11 |
| >5 | 143 | 95 | 42 | 28 | 13 |

For a more intuitive comparison, Figure 11 was drawn. The histogram drawn by the days with MAPE greater than 5% shows that the model performance rankings are GRU, TCN, VMD-TCN, VMD-CNN, and Proposed, respectively. It can also be found in Figure 11 that the addition of VMD has increased the basic performance of the model, that is, the range of the sample MAPE mode up to 2–3%, and the use of CNN has further improved the basic performance of the model to 1–2%. The combination of VMD, CNN, and TCN increases the number of samples in the MAPE range of 2–3% while ensuring that the number of samples in the MAPE range of 1–2% is basically unchanged; it makes the model performance more stable and concentrated.

Figure 12 shows the comparison of each model's prediction results on the day when the proposed model predicts the largest and smallest MAPE, and a day is randomly selected within the MAPE interval where the mode is located to represent the general level of the model. The selected MAPE value is 2.42%. It can be found that the general prediction result of the proposed model is very close to the true value, and the worst prediction result can basically show the trend of load change.

To further verify the generalization of the model in each season, the randomized sample database was counted, and it was found that the number of samples in the spring, summer, autumn, and winter in the test set were 55 days, 55 days, 45 days, and 57 days, respectively. It accounts for 25.94%, 25.94%, 21.22%, and 26.89% of the total number of samples in the test set, as shown in Figure 13a.

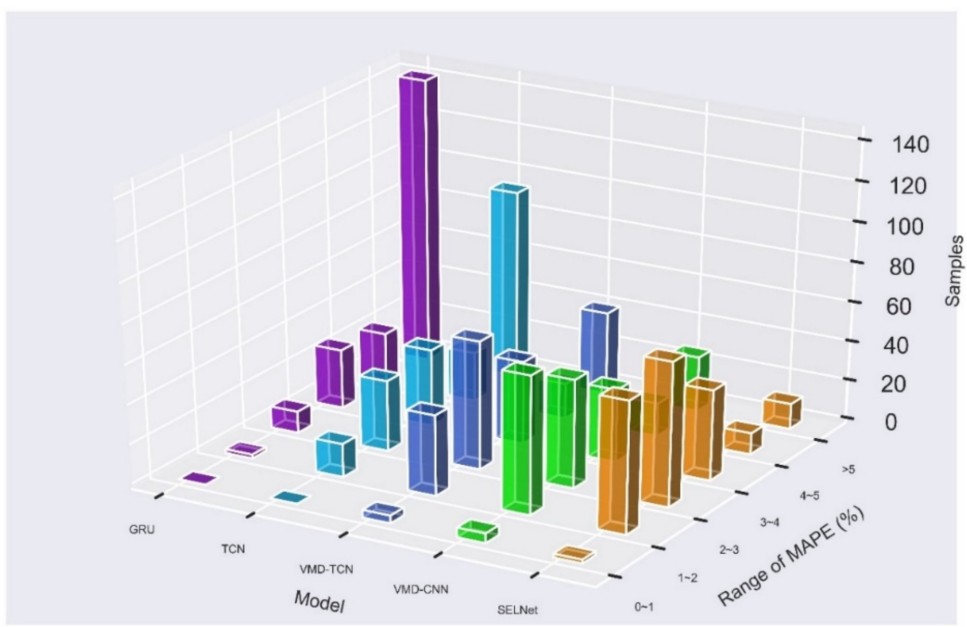

**Figure 11.** The 3D plot of MAPE samples statistics in different ranges for each model.

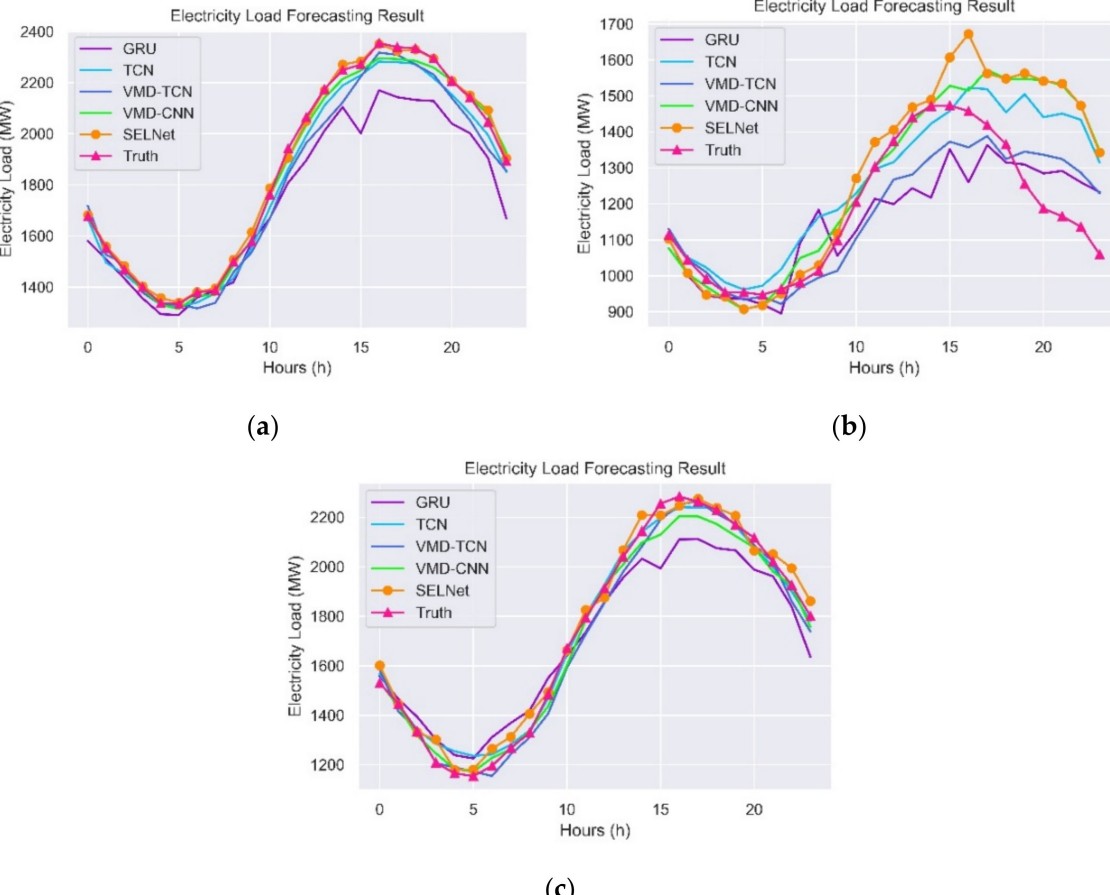

(**a**)

(**b**)

(**c**)

**Figure 12.** (**a**) Comparison of each model of the day when the proposed model predicts the smallest MAPE. (**b**) Comparison of each model of the day when the proposed model predicts the largest MAPE. (**c**) Comparison of each model of the day when the proposed model predicts the general level MAPE.

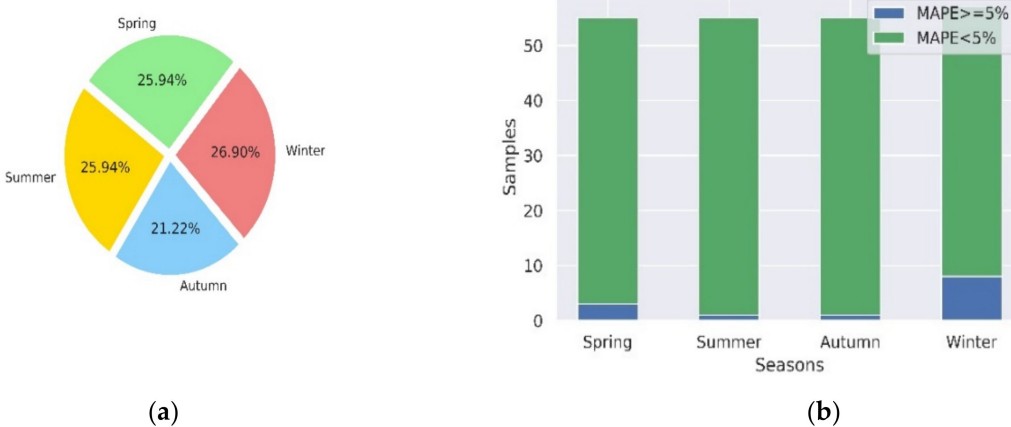

(**a**)                                                                                     (**b**)

**Figure 13.** (**a**) The proportion of each sample in the test set in the four seasons; (**b**) The proportion of MAPE greater than 5% in each sample of the four seasons.

It shows that the samples in each season in the test set occupy a certain proportion. Continue to count the number of samples in each season with a MAPE greater than 5% in the test set. The four seasons are 3, 1, 1, and 8, as shown in Table 6, each accounting for 5.45%, 1.81%, 1.81%, and 14% of the total number of samples in each season in the test set. Figure 13b shows that the number of samples with a MAPE greater than 5% is minimal compared to the total, especially in the three seasons of spring, summer, and autumn, which can be almost ignored, while the performance of the winter model is relatively poor, considering the large difference in winter temperature, There are many festivals, so there are certain changes in residential electricity consumption, which leads to a relatively large number of winter models predicting MAPE greater than 5%.

**Table 6.** MAPE samples statistics for each season.

| Seasons<br>Range (%) | Spring | Summer | Autumn | Winter | Summary |
|---|---|---|---|---|---|
| 0–1 | 0 | 2 | 0 | 0 | 2 |
| 1–2 | 20 | 25 | 14 | 8 | 67 |
| 2–3 | 21 | 11 | 19 | 22 | 73 |
| 3–4 | 8 | 14 | 9 | 15 | 46 |
| 4–5 | 3 | 2 | 2 | 4 | 11 |
| >5 | 3 | 1 | 1 | 8 | 13 |

To show more clearly the prediction performance of the proposed model in four seasons, the number of days in which the MAPE is in each range in the four seasons was counted, as shown in Table 6 and shown in Figure 14. It is not difficult to see that the model performs well in the four seasons. The MAPE is almost completely concentrated in 1–4%, especially 1–3%. The proportion is the largest. Although there are some samples with a MAPE greater than 5% in winter, the proportion is relatively small and does not affect the performance of the model's overall prediction effect. The overall effect is between 1% and 4% of the MAPE, with 2–3% the most. Therefore, the proposed model can make accurate predictions in each season. The best days and worst days for each season in the test set and examples of one-day prediction results in the range of the MAPE mode used to represent the model's general prediction level are given in Figures 15–18 as follows.

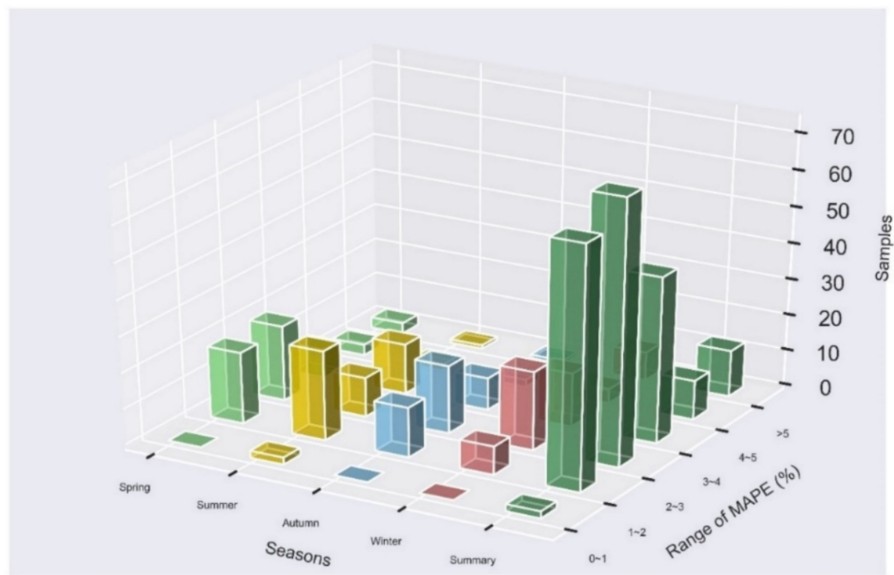

**Figure 14.** The 3D plot of MAPE samples statistics in different ranges in four seasons for SELNet.

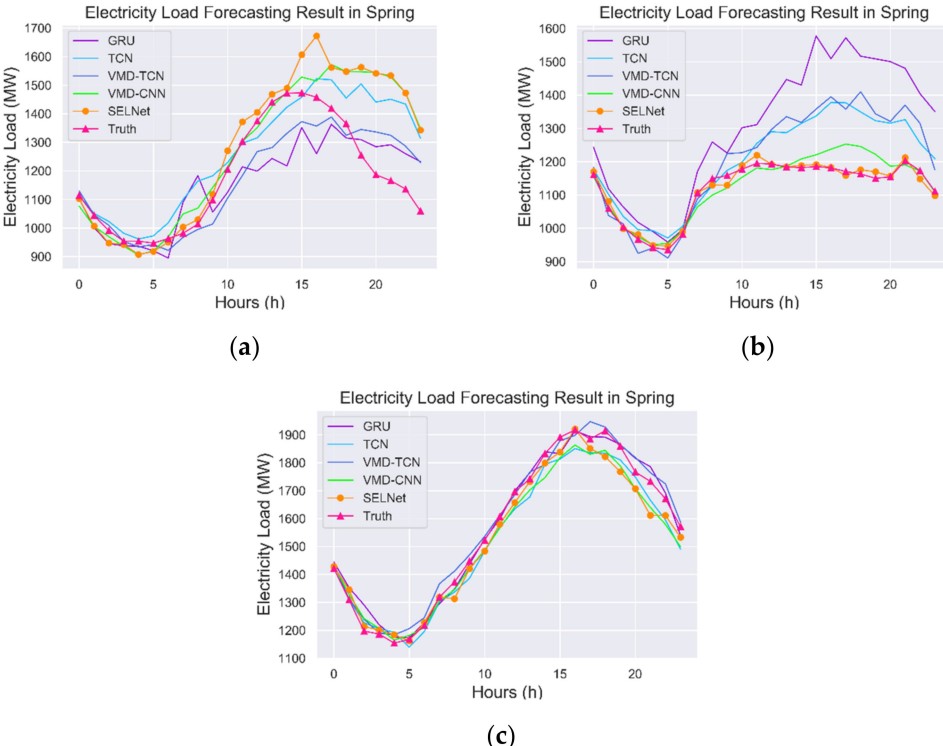

**Figure 15.** (**a**) Comparison for each model of the day in spring when the proposed model predicts the largest MAPE; (**b**) Comparison for each model of the day in spring when the proposed model predicts the smallest MAPE; (**c**) Comparison of each model of the day in spring when the proposed model predicts the general level MAPE.

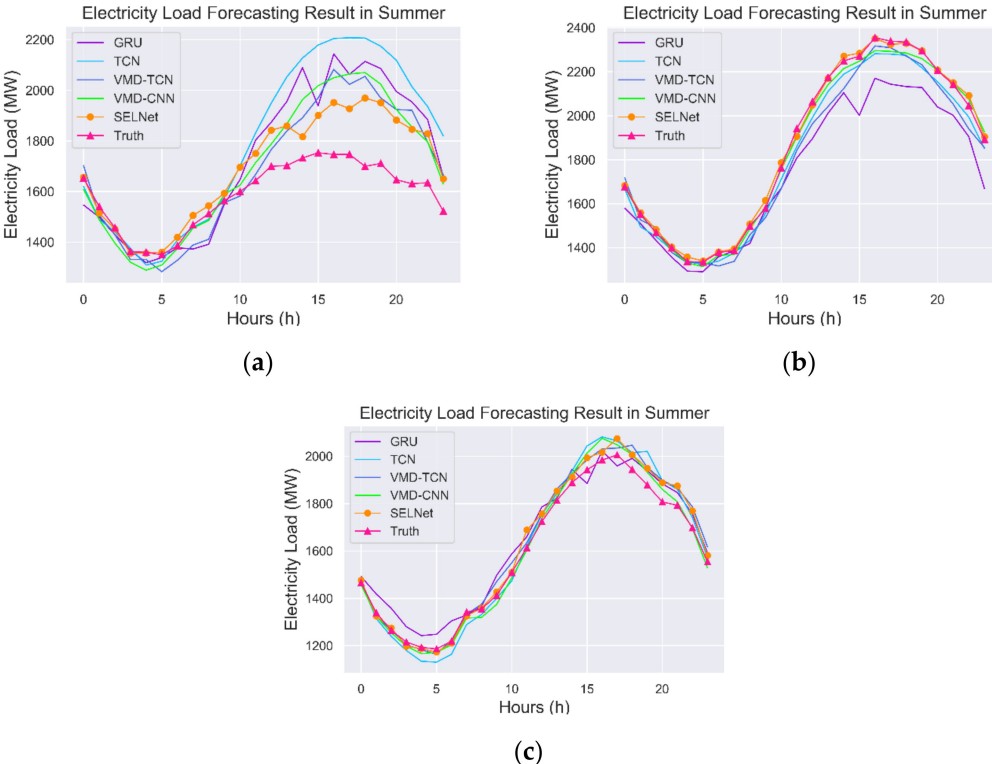

**Figure 16.** (**a**) Comparison for each model of the day in summer when the proposed model predicts the largest MAPE; (**b**) Comparison for each model of the day in summer when the proposed model predicts the smallest MAPE; (**c**) Comparison for each model of the day in summer when the proposed model predicts the general level MAPE.

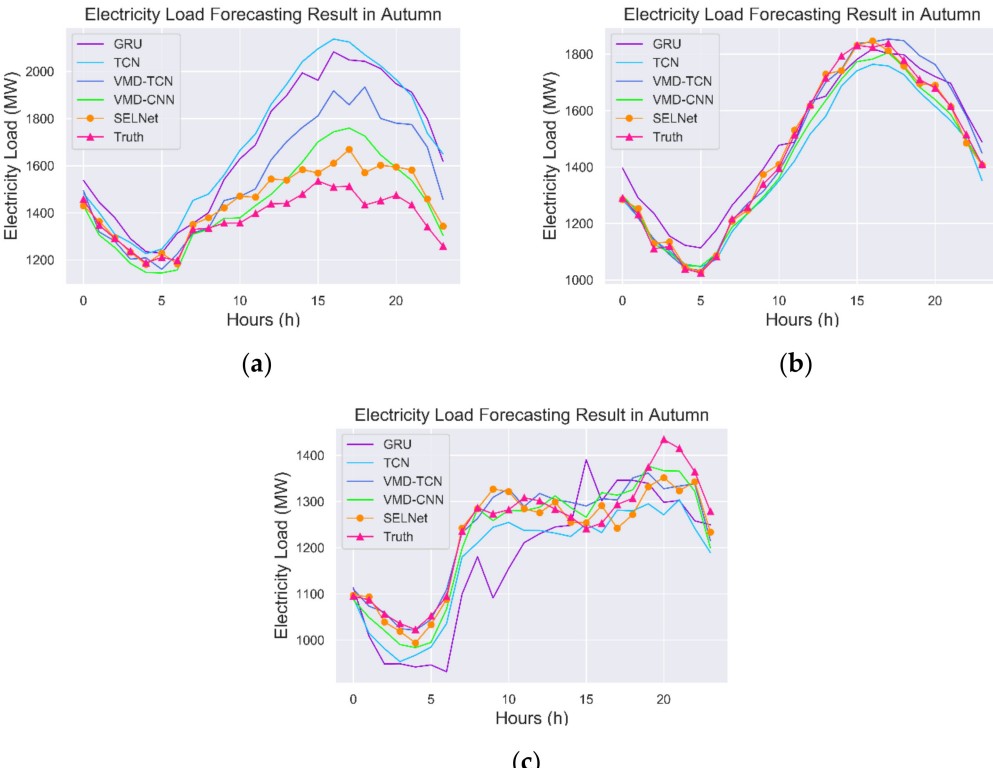

**Figure 17.** (**a**) Comparison for each model of the day in autumn when the proposed model predicts the largest MAPE; (**b**) Comparison for each model of the day in autumn when the proposed model predicts the smallest MAPE; (**c**) Comparison for each model of the day in autumn when the proposed model predicts the general level MAPE.

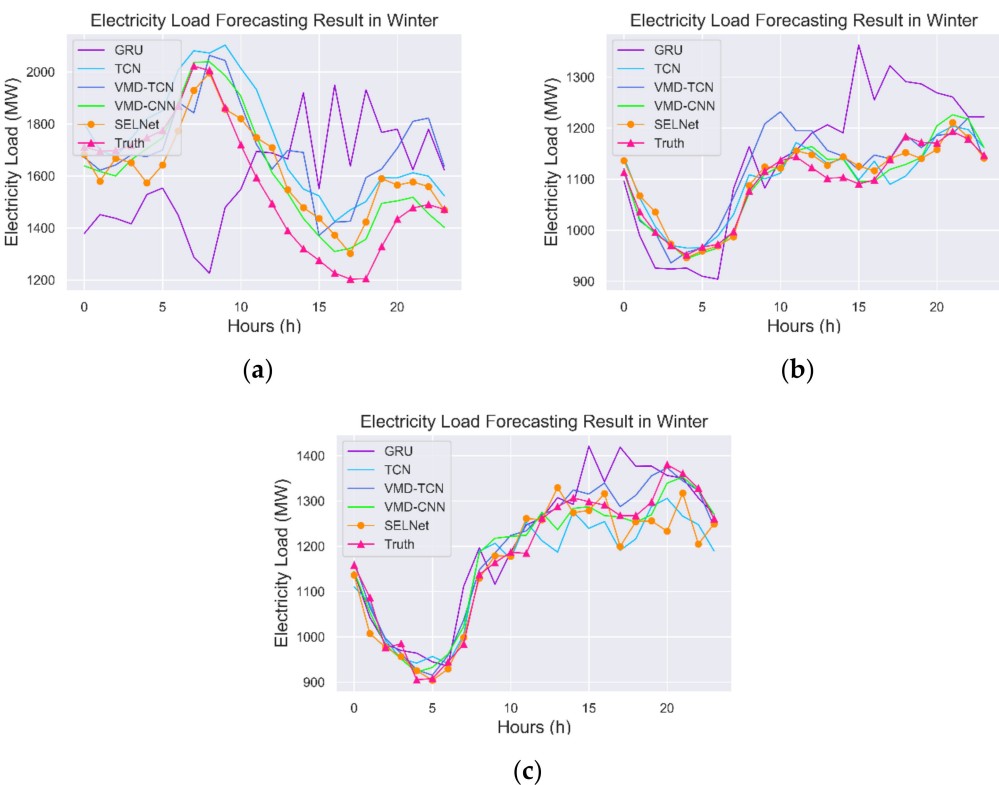

**Figure 18.** (**a**) Comparison for each model of the day in winter when the proposed model predicts the largest MAPE; (**b**) Comparison for each model of the day in winter when the proposed model predicts the smallest MAPE; (**c**). Comparison of each model of the day in winter when the proposed model predicts the general level MAPE.

## 4. Conclusions

Power load forecasting is one of the hottest topics in the electricity market. With the development of distributed generation, the need for grid-connected generation is becoming urgent. It greatly enhances the importance of power load forecasting. At present, many papers have proposed power load forecasting models, but most of the papers have divided the models into seasons or similar days. It will result in a huge amount of model calculation and application complexity. To address this problem, ACF is applied to analyze the selection of input data, VMD is applied to decompose the original load data to a group of relatively stable IMFs. This can greatly reduce the impact of seasonal factors on the model. Some researchers ignore the load time characteristic between the same time, which is a different day. To address this problem, the load date for 168 h a week is expanded into a 24 × 7 matrix, where 24 represents 24 h per day, 7 represents 7 days before the forecast. CNN is applied to extract features from the hourly load on the same day and at the same time load on different days. According to time order, the improved Reshape layer is used to reshape the extracted feature into the matrices like 168 × n, where n represents some channels. Finally, the TCN layer and Dense layer are combined to complete the forecasting. The results of the experiment were verified in the eastern electricity market of Texas. The addition of VMD improved the basic performance of the proposed model, namely sample numbers in the range of MAPE mode, to 2–3%, and the adoption of CNN further improved the model's basic performance to 1–2%. When the combination of VMD, CNN, and TCN can improve, the sample numbers within the 2–3% MAPE are increased while keeping the sample numbers within the 1–2% MAPE range basically unchanged, which makes the performance of the model more stable and centralized. It can also get from an experiment that the proposed model's daily load forecasting performance is outstanding each season.

Above all, the proposed model can be predicting power loads in all seasons, and there is no need to divide four seasons and use four models to predict. This will greatly reduce the utilization of computing equipment, parameters, and computation. The proposed model, which is universally applicable in four seasons, can provide accurate prediction accuracy and help electricity grid managers make better adjustments and deploy power grid operation planning.

Currently, the field of short-term load forecasting is still an essential part of electricity supply and distribution. Table 7 presents a summary of some of the latest research on short-term load forecasting. This table contains the methods and databases used in each study and indicates their performance with different prediction intervals. This table can provide a basic reference for the level of research standards in this domain.

**Table 7.** The latest related studies in the short-term electricity load forecasting domain.

| Authors and Ref | Forecast Horizon | Data Sources | Evaluation Index | Algorithms |
|---|---|---|---|---|
| Wu et al. [42] | One min | Gansu, China | MAPE = 2.8839% | CNN, GRU |
| Jin et al. [43] | One hour | Queensland, Australia | MAPE = 0.7653% | VMD, BEGA, LSTM |
| Nie et al. [44] | One hour | Australia | MAPE = 0.7280% | CEEMD, SSA, RBF, ELM, GRNN |
| Heydari et al. [45] | One hour | America | MAPE = 0.8657% | VMD, GRNN, GSA |
| Shao et al. [46] | Half day | PJM | MAPE = 3.13% | LSTM, CAE, K-means |
| Bedi et al. [47] | One day | Himachal Pradesh, India | MAPE = 3.04% | VMD, ACA, EVM-S, LSTM |
| Deng et al. [48] | One day | Yichun, China | MAPE = 2.057% | VMD, DBN |
| Mansoor et al. [49] | One day | Milan, Italy | MAPE = 2.937% | FFNN, ESN |
| Yin et al. [50] | One day | Guangxi, China | MAPE = 1.89% | MTCN |
| Kong et al. [51] | One day | Tianjin, China | MAPE = 3.104% | DMD, EVCM, SVR |

In further work, the experiment should pay more attention to selecting input and reducing feature redundancy. Features such as temperature can be fused with existing features extracted; this can further reduce the seasons' impact on the model.

**Author Contributions:** Conceptualization: Y.S.; Data curation: Y.M.; Formal analysis: Y.S.; Funding acquisition: P.-H.K. and C.-J.H.; Investigation: C.-J.H.; Methodology: Y.S.; Project administration: P.-H.K.; Resources: Y.M.; Software: Y.S.; Supervision: P.-H.K. and C.-J.H.; Validation: Y.S.; Visualization: Y.S. and Y.M.; Writing—original draft: Y.S., Y.M., S.D., P.-H.K. and C.-J.H.; Writing—review & editing: Y.S., Y.M., S.D. and C.-J.H. All authors have read and agreed to the published version of the manuscript.

**Funding:** This work was supported by the Ministry of Science and Technology, Taiwan, under Grant MOST 109-2221-E-194-053-MY3.

**Institutional Review Board Statement:** Not applicable.

**Informed Consent Statement:** Not applicable.

**Data Availability Statement:** Data available in a publicly accessible repository.

**Conflicts of Interest:** The authors declare no conflict of interest.

## Abbreviations

| | |
|---|---|
| ACF | Autocorrelation function |
| ACA | Agglomerative clustering algorithm |
| AdaBoost | Adaptive boosting |
| AS-GCLSSVM | Autocorrelation feature selection and least squares support vector machine optimizing parameters by grey wolf algorithm and cross validation |
| BEGA | Binary encoding genetic optimization algorithm |
| BPNN | Back propagation neural network |
| Bi-LSTM | Bi-directional LSTM |
| CAE | Convolution autoencoder |

| CNN | Convolutional neural network |
| CV | Cross validation |
| CEEMD | Complementary ensemble empirical mode decomposition |
| DT | Decision tree |
| DMD | Dynamic mode decomposition |
| DBN | Deep belief network |
| EVM-S | Error variance modelling strategy |
| ELM-GA | Extreme learning machine model optimized by genetic algorithm |
| EMD | Empirical mode decomposition |
| ELM | Extreme learning machine |
| EVCM | Extreme value constraint method |
| ESN | Echo state network |
| FFNN | Feed-Forward neural network |
| FA-KELM | kernel extreme learning machine using the firefly algorithm |
| GRU | Gated recurrent unit |
| GWO | Grey wolf optimization algorithm |
| GRNN | Generalized regression neural network |
| GSA | Gravitational search algorithm |
| IMFs | Intrinsic mode functions |
| LSSVM | Least square support vector machine |
| LSTM | Long short-term memory |
| MAPE | Mean absolute percentage error |
| MTCN | Multitemporal-spatial-scale temporal convolutional network |
| RBF | Radial basis function network |
| RMSE | Root mean square error |
| RNN | Recurrent neural network |
| SSA | Singular spectrum analysis |
| SVR | Support vector regression |
| TCN | Temporal convolutional network |
| Uni-LSTM | Uni-directional LSTM |
| VMD | Variational mode decomposition |
| VMD-CNN | Variational mode decomposition and convolutional neural network |
| VMD-TCN | Variational mode decomposition and temporal convolutional network |
| XGBoost | Extreme gradient boosting model |
| 1D-CNN | One-dimensional convolutional neural network |
| 2D-CNN | Two-dimensional convolutional neural network |

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
