# Peer review of "An Ensemble Model based on Deep Learning and Data Preprocessing for Short-Term Electrical Load Forecasting"

_sustainability, doi:10.3390/su13041694_

Round 1

Reviewer 1 Report

This manuscript proposes a new method, called SELNet, to address the electrical load forecasting. This method is composed of various algorithms at different stages, such as the variational mode decomposition (VMD) algorithm to transform the input data into intrinsic mode functions (IMFs), a 2D convolutional neural network (2D-CNN) for features extraction, and the temporal convolutional network (TCN) algorithm for the final prediction.

The manuscript provides a comprehensive description of the methods being employed.

The dataset used is presented adequately and offer a clear overall picture. The practical treatment regarding the proposed methods is fairly well explained and seems correct, and I believe it represents a good contribution to this area of research.

However, when the predictive performance of the constituent parts (VMD-TCN and VMD-CNN methods) of the final proposed method are included in the practical experiments, this creates a potential confusion for the readers. For instance, I had to peruse the manuscript numerous times in order to clarify wether the VMD-TCN and VMD-CNN methods are actually components of your proposed method or used previously by other researchers.

Therefore, this aspect must be addressed more clearly throughout the manuscript to alleviate any potential confusion. I would suggest first to clarify this aspect in the abstract. Even though this aspect is already addressed, it is not clearly expressed. Also, state it clearly in the “main contributions” paragraph and in the “Proposed model” section. I would use more frequently the “SELNet” name and state it clearly that the VMD-TCN and VMD-CNN methods are its components. I would also replace “Proposed” in every table and figure with “SELNet” — this would improve clarity further.

Other minor corrections to be addressed:

  • Please explain each abbreviation first used in the manuscript — for instance in the abstract, GRU is used but not explained.
  • At page 11, line 313 — Please revise “…quartile Figure, as shown in Figure 4.”
  • At page 11, line 334 — Please revise “The importance of TCN for time series learning.”
  • At page 15, line 400 — Please revise — “TCN and Dense are used to complete the forecasting.”
  • Please make sure to end each sentence with a dot. There are many instances where you do not have one.

Author Response

This manuscript proposes a new method, called SELNet, to address electrical load forecasting. This method is composed of various algorithms at different stages, such as the variational mode decomposition (VMD) algorithm to transform the input data into intrinsic mode functions (IMFs), a 2D convolutional neural network (2D-CNN) for features extraction, and the temporal convolutional network (TCN) algorithm for the final prediction. The manuscript provides a comprehensive description of the methods being employed. The dataset used is presented adequately and offers a clear overall picture. The practical treatment regarding the proposed methods is fairly well explained and seems correct, and I believe it represents a good contribution to this area of research.

However, when the predictive performance of the constituent parts (VMD-TCN and VMD-CNN methods) of the final proposed method are included in the practical experiments, this creates potential confusion for the readers. For instance, I had to peruse the manuscript numerous times in order to clarify whether the VMD-TCN and VMD-CNN methods are actually components of your proposed method or used previously by other researchers.

Therefore, this aspect must be addressed more clearly throughout the manuscript to alleviate any potential confusion. I would suggest first clarifying this aspect in the abstract. Even though this aspect is already addressed, it is not clearly expressed. Also, state it clearly in the “main contributions” paragraph and in the “Proposed model” section. I would use more frequently the “SELNet” name and state it clearly that the VMD-TCN and VMD-CNN methods are its components. I would also replace “Proposed” in every table and figure with “SELNet” — this would improve clarity further.

Other minor corrections to be addressed:

  • Please explain each abbreviation first used in the manuscript — for instance in the abstract, GRU is used but not explained.
  • At page 11, line 313 — Please revise “…quartile Figure, as shown in Figure 4.”
  • At page 11, line 334 — Please revise “The importance of TCN for time series learning.”
  • At page 15, line 400 — Please revise — “TCN and Dense are used to complete the forecasting.”
  • Please make sure to end each sentence with a dot. There are many instances where you do not have one.
  • (Reviewer 1) Question 1: However, when the predictive performance of the constituent parts (VMD-TCN and VMD-CNN methods) of the final proposed method are included in the practical experiments, this creates a potential confusion for the readers. For instance, I had to peruse the manuscript numerous times in order to clarify whether the VMD-TCN and VMD-CNN methods are actually components of your proposed method or used previously by other researchers. Therefore, this aspect must be addressed more clearly throughout the manuscript to alleviate any potential confusion.

(Author) Respond:

 Yes, the architectures of VMD-CNN and VMD-TCN were found using a grid search approach. To highlight the advantages of the proposed model when combined with VMD, 2D-CNN, and TCN, we used VMD-TCN and VMD-CNN to forecast the electricity load. We added the following sentence to highlight the advantages of the model:

On page 1, line 31 — “The TCN exhibited better performance than the GRU in load forecasting. The mean absolute percentage error (MAPE) of the TCN, which was over 5%, was less than that of the GRU. Following the addition of VMD to the TCN, the basic performance of the model was 2%–3%. A comparison between the SELNet model and the VMD-TCN model indicated that the application of a 2D-CNN improves the forecast performance, with only a few samples having an MAPE of over 4%.”                

Hence, the VMD-TCN and VMD-CNN models help us understand the manner in which the proposed model impact data processing and load forecasting.

To eliminate any potential confusion, we have included the following statements:

On page 1, line 32 — “To demonstrate the effectiveness of the proposed model for each part, we compared it with the gated recurrent unit (GRU), TCN, VMD-TCN, and VMD-CNN models.”

On page 3, line 147 — “This study combines VMD, CNN, and TCN to forecast next-day hourly electricity load data. To verify the effectiveness of SELNet, we compared it with the GRU, TCN, VMD-TCN, and VMD-CNN models; the results are discussed in Section 3.”

On page 8, line 287— “The addition of VMD, CNN, and TCN enhances the performance of SELNet; this is discussed in Section 3.”

Some studies have employed VMD-CNN to achieve accurate time-series forecasting:

  1. Short-Term PV Power Forecasting for Renewable Energy Using Hybrid Spider Optimization-Based Convolutional Neural Network [1].
  2. An Early Fault Diagnosis Method Based on the Optimization of a Variational Modal Decomposition and Convolutional Neural Network for Aeronautical Hydraulic Pipe Clamps [2].
  3. Multiple Sensors Fault Diagnosis for Rolling Bearing Based on Variational Mode Decomposition and Convolutional Neural Networks [3].
  4. Risk Forecasting in the Crude Oil Market: A Multiscale Convolutional Neural Network Approach [4].

These studies [1]–[4] mainly employed a hybrid VMD-CNN model to develop a time-series database and extract features. The VMD-CNN is also a classic hybrid model and one that has undergone development in recent years. Because CNN's have strong adaptability in the deep learning domain, we considered a classic hybrid model “VMD-CNN” for comparison, with proper justification. Please refer to the following papers [1]–[4].

  1. Ghosh, D. Short-Term PV Power Forecasting for Renewable Energy Using Hybrid Spider Optimization-Based Convolutional Neural Network. In Innovative Product Design and Intelligent Manufacturing 1. Ghosh, D. Short-Term PV Power Forecasting for Renewable Energy Using Hybrid Spider Optimization-Based Convolutional Neural Network. In Innovative Product Design and Intelligent Manufacturing Systems.; 2020; pp. 809–817 ISBN 9789811526954.
  2. Yang, T.; Yu, X.; Li, G.; Dou, J.; Duan, B. An early fault diagnosis method based on the optimization of a variational modal decomposition and convolutional neural network for aeronautical hydraulic pipe clamps. Meas. Sci. Technol. 2020, 31, 055007, doi:10.1088/1361-6501/ab5342.
  3. Hou, Q.; Wang, J.; Shen, Y. Multiple Sensors Fault Diagnosis for Rolling Bearing Based on Variational Mode Decomposition and Convolutional Neural Networks. In Proceedings of the 2020 11th International Conference on Prognostics and System Health Management (PHM-2020 Jinan); IEEE, 2020; pp. 450–455.
  4. Zou, Y.; Yu, L.; Tso, G.K.F.; He, K. Risk forecasting in the crude oil market: A multiscale Convolutional Neural Network approach. Phys. A Stat. Mech. its Appl. 2020, 541, 123360, doi:10.1016/j.physa.2019.123360.Systems.; 2020; pp. 809–817 ISBN 9789811526954.

We also provide a table to show the novelty model for load forecasting and the performance of these model are also give in the table. Please see the following table.

Authors and Ref

Forecast horizon

Data sources

Outcome

Algorithms

Wu et al.

One min

Gansu,

China

MAPE=2.8839%

CNN, GRU

Jin et al.

One hour

Queensland,

Australia

MAPE=0.7653%

VMD, BEGA, LSTM

Nie et al.

One hour

Australia

MAPE= 0.7280%

CEEMD, SSA, RBF, ELM, GRNN

Heydari et al.

One hour

America

MAPE= 0.8657%

VMD, GRNN, GSA

Shao et al.

Half day

PJM

MAPE=3.13%

LSTM, CAE,

K-means

Bedi et al.

One day

Himachal Pradesh, India

MAPE=3.04%

VMD, AE, EVM-S, LSTM

Deng et al.

One day

Yichun, China

MAPE=2.057%

VMD, DBN

Mansoor et al.

One day

Milan, Italy

MAPE=2.937%

FFNN, ESN

Yin et al.

One day

Guangxi, China

MAPE=1.89%

MTCN

Kong et al.

One day

Tianjin, China

MAPE=3.104%

DMD, EVCM, SVR

  • (Reviewer 1) Question 2: I would suggest first to clarify this aspect in the abstract. Even though this aspect is already addressed, it is not clearly expressed. Also, state it clearly in the “main contributions” paragraph and in the “Proposed model” section.

(Author) Respond:

We have clarified this aspect in the abstract as well as in the main text.

On page 1, line 27 — “To demonstrate the effectiveness of the proposed model for each part, we compared it with the gated recurrent unit (GRU), TCN, VMD-TCN, and VMD-CNN models.”

On page 3, line 144 — “This study combines VMD, CNN, and TCN to forecast next-day hourly electricity load data. To verify the effectiveness of SELNet, we compared it with the GRU, TCN, VMD-TCN, and VMD-CNN models; the results are discussed in Section 3.”

On page 8, line 280 — “The addition of VMD, CNN, and TCN enhances the performance of SELNet; this is discussed in Section 3.”

  • (Reviewer 1) Question 3: I would use more frequently the “SELNet” name and state it clearly that the VMD-TCN and VMD-CNN methods are its components. I would also replace “Proposed” in every table and figure with “SELNet” — this would improve clarity further.

(Author) Respond:

VMD-TCN and VMD-CNN are the models that we selected for comparison with our proposed model SELNet, which incorporates VMD, CNN, and TCN. We have clarified the use of VMD-TCN and VMD-CNN in our responses to questions 1 and 2.

We have replaced “Proposed” in every table and figure with “SELNet.” Please see the revised manuscript.

  • (Reviewer 1) Question 4: Please explain each abbreviation first used in the manuscript — for instance in the abstract, GRU is used but not explained.

(Author) Respond:

We have rechecked the paper and defined each abbreviation at its first occurrence in the manuscript.

  • (Reviewer 1) Question 5: At page 11, line 313 — Please revise “…quartile Figure, as shown in Figure 4.”

(Author) Respond:

We have reviewed the paper and resolved this issue.

  • (Reviewer 1) Question 6: At page 11, line 334 — Please revise “The importance of TCN for time series learning.”

(Author) Respond:

We have revised the description as follows:

On page 12, line 373 — “Therefore, using a CNN to extract features from the expanded matrices, reshaping them into a time series, and applying a TCN to learn the time series and realize the final forecasting is an effective approach.”

  • (Reviewer 1) Question 7: At page 15, line 400 — Please revise — “TCN and Dense are used to complete the forecasting.”

(Author) Respond:

 We have revised the description as follows:

 On page 15, line 439 —“the TCN layer and Dense layer are combined to complete the forecasting process.”

  • (Reviewer 1) Question 8: Please make sure to end each sentence with a dot. There are many instances where you do not have one.

(Author) Respond:

We have reviewed the paper and resolved this issue.

Reviewer 2 Report

This work describes an ensemble model for short term load forecasting based a CNN which receives data previously processed by a Variational Mode Decomposition. The model provides good results which are well describe in the text. Nevertheless the manuscript needs an in depth revision to justify its publication. General comments are provided bellow followed by some particular ones.

Acronyms must be explained the first time they are used.

“1. Introduction”. It is not clear and does not provide an overview of the techniques and tools other authors have used to deal with the problem of short term load forecasting.

The text presents a description of the works that several authors have carried out but neither presents a classification nor a description of pros and cons of the different forecasting tools that can be used in short term load forecasting. They should provide this classification. For example, they could describe “classical “widely tools used such as ARIMA pointing out their limits, followed by a study of those based on Artificial Intelligent methods, concluding with those the authors have selected in their work.

The description of the work carried out in the paper should be clearer and more concise. It should be described in depth in a later section.

The author should clearly explain why they have selected the tools they use in their work and no others.

“2.1 Variational Mode Decomposition”. The description of this method is neither clear nor provides an idea of how it works. It should be rewritten describing how the method works.

“2.2 Two-Dimensional Convolutional Neural Network”. The description of this structure is very poor and does not help to understand how a CNN works. It should be rewritten describing the structure of the network and how the convolutional process is carried out by that structure. The concept of filters may be also explained by linking it to the convolutional kernel.

“2.3.2. Dilated Convolution” The kernel structure used to carry out the convolution process is not clear. The difference with “Causal convolution” should be further explained.

“2.3.3. Residual Block” Should be rewritten. A reader who has no previous knowledge of this structure will not understand how it works.

“2.3.4. Proposed Model” The authors should further explain and justify the model they propose. They should justify why they have selected three convolutional layers, and that number of filters. Why they use a TCN structure after the convolutional layers? How the TCN works? Why they define a 1-D layer after TCN? What does this layer do? How the outputs of this layer are comprised to fir the 24x1 output layer structure? Why the inputs to the model are 1057 samples (Table 1 and Fig 5) when this is the total amount of samples? It should be clarified which is the number and structure of the input data presented to the model each time it is run. The training process should also been explained.

It should be clarified in the text that the Variational Mode Decomposition performs a data preprocessing stage in the model.

“4. Conclusion”. The authors should provide a comparison with the performances achieved in other works where hourly load was forecasted in order to prove that the model they have presented performs better than other models.

"References". The authors should provide further references related to short term load forecasting, mainly those using artificial intelligence methods, as one of them is used in this work. They should also provide basic references where Variational Mode Decompsition and CNN are studied.

Lines 27: What do “sample number” and MAPE mode” mean?

Lines 27-30: The description of the results achieved should be rewritten. It is not clear what the authors mean.

Line 54: The expression “length of the forecast time” is not clear, “forecasting horizon” would be better.

Line 72: The meaning of CNN should be explained.

Line 75: The meaning of AS-GCLSSVM should be explained.

Line 77: The meaning of GWO and CV should be explained.

Lines 85.86: “For some nonlinear systems, the fuzzy neural 85 network also needs to be combined with Fourier and other regression models” Some references should be provided to support this statement.

Line 77: The meaning of BPNN should be explained.

Line 93: The meaning of ELM-GA should be explained.

Line 104: The meaning of FA-KELM should be explained.

Line 115: The meaning of 2D-CNN should be explained.

Line 127: The meaning of TCN should be explained.

Line 130: The meaning of GRU should be explained.

Line 181: the concept of residual linking should be further explained.

Lines 227-229: How have the structure of kernel, filters and output data been defined? The authors should justify why they have used those values. They should also clarity what a filter is in this structure.

Lines 238-239: What do the authors mean with “…the same time of the week…”?

Line 284: Which is the difference between “...data for five weeks...” and “...electricity load data...“

Line 285: It should be Figure 9 instead of Figure 3.

Line 313: It should be Figure 10 instead of Figure 4.

Author Response

This work describes an ensemble model for short-term load forecasting based a CNN which receives data previously processed by a Variational Mode Decomposition. The model provides good results which are well described in the text. Nevertheless, the manuscript needs an in depth revision to justify its publication. General comments are provided below followed by some particular ones.

Acronyms must be explained the first time they are used.

“1. Introduction”. It is not clear and does not provide an overview of the techniques and tools other authors have used to deal with the problem of short term load forecasting. The text presents a description of the works that several authors have carried out but neither presents a classification nor a description of the pros and cons of the different forecasting tools that can be used in short-term load forecasting. They should provide this classification. For example, they could describe “classical “widely tools used such as ARIMA pointing out their limits, followed by a study of those based on Artificial Intelligent methods, concluding with those the authors have selected in their work.

The description of the work carried out in the paper should be clearer and more concise. It should be described in-depth in a later section. The author should clearly explain why they have selected the tools they use in their work and no others.

“2.1 Variational Mode Decomposition”. The description of this method is neither clear nor provides an idea of how it works. It should be rewritten to describe how the method works.

“2.2 Two-Dimensional Convolutional Neural Network”. The description of this structure is very poor and does not help to understand how a CNN works. It should be rewritten to describe the structure of the network and how the convolutional process is carried out by that structure. The concept of filters may be also explained by linking it to the convolutional kernel.

“2.3.2. Dilated Convolution” The kernel structure used to carry out the convolution process is not clear. The difference with “Causal convolution” should be further explained.

“2.3.3. Residual Block” Should be rewritten. A reader who has no previous knowledge of this structure will not understand how it works.

“2.3.4. Proposed Model” The authors should further explain and justify the model they propose. They should justify why they have selected three convolutional layers and that number of filters. Why they use a TCN structure after the convolutional layers? How the TCN works? Why they define a 1-D layer after TCN? What does this layer do? How the outputs of this layer are comprised to fir the 24x1 output layer structure? Why the inputs to the model are 1057 samples (Table 1 and Fig 5) when this is the total amount of samples? It should be clarified which is the number and structure of the input data presented to the model each time it is run. The training process should also been explained.

It should be clarified in the text that the Variational Mode Decomposition performs a data preprocessing stage in the model.

“4. Conclusion”. The authors should provide a comparison with the performances achieved in other works where hourly load was forecasted in order to prove that the model they have presented performs better than other models.

"References". The authors should provide further references related to short term load forecasting, mainly those using artificial intelligence methods, as one of them is used in this work. They should also provide basic references where Variational Mode Decompsition and CNN are studied.

Lines 27: What do “sample number” and MAPE mode” mean?

Lines 27-30: The description of the results achieved should be rewritten. It is not clear what the authors mean.

Line 54: The expression “length of the forecast time” is not clear, “forecasting horizon” would be better.

Line 72: The meaning of CNN should be explained.

Line 75: The meaning of AS-GCLSSVM should be explained.

Line 77: The meaning of GWO and CV should be explained.

Lines 85.86: “For some nonlinear systems, the fuzzy neural 85 network also needs to be combined with Fourier and other regression models” Some references should be provided to support this statement.

Line 77: The meaning of BPNN should be explained.

Line 93: The meaning of ELM-GA should be explained.

Line 104: The meaning of FA-KELM should be explained.

Line 115: The meaning of 2D-CNN should be explained.

Line 127: The meaning of TCN should be explained.

Line 130: The meaning of GRU should be explained.

Line 181: the concept of residual linking should be further explained.

Lines 227-229: How have the structure of kernel, filters and output data been defined? The authors should justify why they have used those values. They should also clarity what a filter is in this structure.

Lines 238-239: What do the authors mean with “…the same time of the week…”?

Line 284: Which is the difference between “...data for five weeks...” and “...electricity load data...“

Line 285: It should be Figure 9 instead of Figure 3.

Line 313: It should be Figure 10 instead of Figure 4.

  • (Reviewer 2) Question 1: Acronyms must be explained the first time they are used.

(Author) Respond:

Thank you for your comment. We have reviewed the paper and defined each abbreviation at its first occurrence in the manuscript. We have also provided a list of abbreviations in “Appendix A.”

  • (Reviewer 2) Question 2: It is not clear and does not provide an overview of the techniques and tools other authors have used to deal with the problem of short term load forecasting. The text presents a description of the works that several authors have carried out but neither presents a classification nor a description of the pros and cons of the different forecasting tools that can be used in short-term load forecasting. They should provide this classification. For example, they could describe “classical “widely tools used such as ARIMA pointing out their limits, followed by a study of those based on Artificial Intelligent methods, concluding with those the authors have selected in their work.

(Author) Respond:

We have reviewed the Introduction section and made relevant changes. We have described electrical load forecasting as a process of “Regression Modeling–Machine Learning–Deep Learning.”

  • (Reviewer 2) Question 3: The description of the work carried out in the paper should be clearer and more concise. It should be described in depth in a later section. The author should clearly explain why they have selected the tools they use in their work and no others.

(Author) Respond:

Thank you for your comment. We have described the advantages of each part comprising SELNet.

On page 4, line 152 — “Unlike classic EMD [23], VMD uses a nonrecursive form to complete signal decomposition, wherein the decomposed subsignals are extracted simultaneously. VMD increases the bandwidth limitations and reconstruction fidelity constraints during signal decomposition, and therefore, it is more robust in terms of noise sensitivity. VMD determines the mode and the corresponding center frequency set according to the constraint conditions to reconstruct the original signal.”
    Because TCN uses causal convolution, there is a causal relationship between convolutional network layers and layers. It is a one-way structure with strict time constraints.

On page 5, line 202 — “The effective window of the dilated convolution will increase exponentially as the sampling rate of the dilated convolution increases. Therefore, the TCN can use fewer layers to obtain a large receptive field.

Residual linking has been proven to be an effective method for training deep networks. It allows the network to transmit information in a cross-layer manner, reduces problems such as gradient disappearance, and enhances the model’s robustness.”

On page 7, line 255 — “Using this model to decompose the original time-series signal ensures that the seasonal variation trend in power load has negligible influence on the prediction accuracy of the model; this considerably reduces the complexity of the model. As a new signal decomposition method, VMD performs better than classic EMD. The model uses the classic and effective CNN to extract features between different days and the same time period to increase the prediction accuracy. Compared with other models, TCN exhibits better performance in time-series forecasting. The TCN model can not only contain more historical features but also reduce the amount of calculation considerably.”

  • (Reviewer 2) Question 4: “2.1 Variational Mode Decomposition” The description of this method is neither clear nor provides an idea of how it works. It should be rewritten describing how the method works.

(Author) Respond:

We have rewritten the description of this method and added details regarding its working mechanism.

  • (Reviewer 2) Question 5: “2.2 Two-Dimensional Convolutional Neural Network”. The description of this structure is very poor and does not help to understand how a CNN works. It should be rewritten describing the structure of the network and how the convolutional process is carried out by that structure. The concept of filters may be also explained by linking it to the convolutional kernel.

(Author) Respond:

To help readers understand the working mechanism of the CNN, we have made necessary revisions to the manuscript.

  • (Reviewer 2) Question 6: “2.3.2. Dilated Convolution” The kernel structure used to carry out the convolution process is not clear. The difference with “Causal convolution” should be further explained.

(Author) Respond:

A description of the kernel structure is provided on line 215: “The size of the convolution kernel is 2.”

Some content has been added to highlight the difference between dilated convolution and causal convolution.

On page 6, line 232 — “The difference between dilated convolution and causal convolution is the addition of dilatation coefficients 1 and 2 to dilated convolution, which expands the receptive field of y1 to x4 and reduces the amount of calculation.”

  • (Reviewer 2) Question 7:“2.3.3. Residual Block” Should be rewritten. A reader who has no previous knowledge of this structure will not understand how it works.

(Author) Respond:

Some content has been added to the text to explain this structure in detail.

On page 6, line 239 — “Therefore, the results of shallow network training may be better than those of deep network training. To solve this problem, the shallow network is identically mapped to the deep network and the residual characteristics are constantly updated to ensure that the training effect of the deep network is not worse than that of the shallow network.”

  • (Reviewer 2) Question 8: “2.3.4. Proposed Model” The authors should further explain and justify the model they propose. They should justify why they have selected three convolutional layers, and that number of filters. Why they use a TCN structure after the convolutional layers? How the TCN works? Why they define a 1-D layer after TCN? What does this layer do? How the outputs of this layer are comprised to fir the 24x1 output layer structure? Why the inputs to the model are 1057 samples (Table 1 and Fig 5) when this is the total amount of samples? It should be clarified which is the number and structure of the input data presented to the model each time it is run. The training process should also been explained.

(Author) Respond:

Although recent research has highlighted some hot topics regarding neural architecture search and provided explanations of the model, deep learning architectures are widely set through experiments. Hence, we employed a grid search approach to identify the architecture that exhibits the best performance.

TCN is widely known for its outstanding performance in time-series forecasting. It is used to process time-series data such as RNN, LSTM, and GRU. The TCN has some of the following advantages.

Because the TCN uses causal convolution, there exists a causal relationship between convolutional network layers and layers. It is a one-way structure with strict time constraints.

The effective window of the dilated convolution increases exponentially as the sampling rate of the dilated convolution increases. Therefore, the TCN can use fewer layers to obtain a large receptive field.

Residual linking has been proven to be an effective method for training deep networks. It allows the network to transmit information in a cross-layer manner, reduces problems such as gradient disappearance, and enhances the model’s robustness.

A 1D-layer is used to process all the time-series information extracted by the TCN and output the forecast result. We forecast the next-day hourly load; hence, the output is .

The data is divided into 1057 samples, which are used to train and test the network; we have discussed this on page 9, line 288. Perhaps Figure 5 and Table 1 caused some confusion; we have revised them accordingly.

 The input shape is constant and does not change the weight of trained parameters. The training process is performed using error back-propagation. Because we deemed the explanation unnecessary for our paper, and because most studies do not explain it, we did not describe it. Having said that, we acknowledge your comment, and we will pay more attention to this in future experiments. Taking your advice, we have revised Table 1 and Figure 5. We have described our model more clearly and have highlighted the advantages of each basic model that we selected to construct SELNet.

  • (Reviewer 2) Question 9: It should be clarified in the text that the Variational Mode Decomposition performs a data preprocessing stage in the model.

(Author) Respond:

We have mentioned that the experiment consists of two parts: data processing and load forecasting (see page 1, line 19).

  • (Reviewer 2) Question 10: “4. Conclusion”. The authors should provide a comparison with the performances achieved in other works where hourly load was forecasted in order to prove that the model they have presented performs better than other models.

(Author) Respond:

We have summarized the latest research on short-term electricity load forecasting models in a table. The table includes all models used and their performance.

On page 16, line 453 — “Currently, the field of short-term load forecasting is still an essential part of electricity supply and distribution. Table 7 presents a summary of some of the latest research on short-term load forecasting. This table contains the methods and databases used in each study and indicates their performance with different prediction intervals. This table can provide a basic reference for the level of research standards in this domain.”

Table 7. The latest related researches in short-term electricity load forecasting domain.

Authors and Ref

Forecast horizon

Data sources

Outcome

Algorithms

Wu et al.

One min

Gansu,

China

MAPE=2.8839%

CNN, GRU

Jin et al.

One hour

Queensland,

Australia

MAPE=0.7653%

VMD, BEGA, LSTM

Nie et al.

One hour

Australia

MAPE= 0.7280%

CEEMD, SSA, RBF, ELM, GRNN

Heydari et al.

One hour

America

MAPE= 0.8657%

VMD, GRNN, GSA

Shao et al.

Half day

PJM

MAPE=3.13%

LSTM, CAE,

K-means

Bedi et al.

One day

Himachal Pradesh, India

MAPE=3.04%

VMD, AE, EVM-S, LSTM

Deng et al.

One day

Yichun, China

MAPE=2.057%

VMD, DBN

Mansoor et al.

One day

Milan, Italy

MAPE=2.937%

FFNN, ESN

Yin et al.

One day

Guangxi, China

MAPE=1.89%

MTCN

Kong et al.

One day

Tianjin, China

MAPE=3.104%

DMD, EVCM, SVR

  • (Reviewer 2) Question 10: "References". The authors should provide further references related to short term load forecasting, mainly those using artificial intelligence methods, as one of them is used in this work. They should also provide basic references where Variational Mode Decompsition and CNN are studied.

(Author) Respond:

We have cited several recent papers that have used VMD, CNN, or the hybrid VMD-CNN model for time-series forecasting. Please refer to the table in the previous response. These related works demonstrate that many researchers believe that the VMD and CNN models are useful for time-series forecasting. Hence, we selected the classic VMD-CNN model for comparison in our study. In addition, other deep learning models such as the GRU and LSTM are cited in the table. The GRU and LSTM models exhibit good performance in time-series processing; therefore, we cited some studies that have used the LSTM and GRU models for comparison.

The following studies have used VMD-CNN to achieve accurate time-series forecasting:

  1. Short-Term PV Power Forecasting for Renewable Energy Using Hybrid Spider Optimization-Based Convolutional Neural Network [1].
  2. An Early Fault Diagnosis Method Based on the Optimization of a Variational Modal Decomposition and Convolutional Neural Network for Aeronautical Hydraulic Pipe Clamps [2].
  3. Multiple Sensors Fault Diagnosis for Rolling Bearing Based on Variational Mode Decomposition and Convolutional Neural Networks [3].
  4. Risk Forecasting in the Crude Oil Market: A Multiscale Convolutional Neural Network Approach [4].

These studies [1]–[4] mainly employed the hybrid VMD-CNN model to develop a time-series data and extract features. The VMD-CNN is also a classic hybrid model and one that has recently undergone development. Because CNNs have favorable adaptability for the deep learning domain, we considered a classic hybrid model “VMD-CNN” for the comparison, with proper justification. Please refer to the following papers [1]–[4].

  1. Ghosh, D. Short-Term PV Power Forecasting for Renewable Energy Using Hybrid Spider Optimization-Based Convolutional Neural Network. In Innovative Product Design and Intelligent Manufacturing 1. Ghosh, D. Short-Term PV Power Forecasting for Renewable Energy Using Hybrid Spider Optimization-Based Convolutional Neural Network. In Innovative Product Design and Intelligent Manufacturing Systems.; 2020; pp. 809–817 ISBN 9789811526954.
  2. Yang, T.; Yu, X.; Li, G.; Dou, J.; Duan, B. An early fault diagnosis method based on the optimization of a variational modal decomposition and convolutional neural network for aeronautical hydraulic pipe clamps. Meas. Sci. Technol. 2020, 31, 055007, doi:10.1088/1361-6501/ab5342.
  3. Hou, Q.; Wang, J.; Shen, Y. Multiple Sensors Fault Diagnosis for Rolling Bearing Based on Variational Mode Decomposition and Convolutional Neural Networks. In Proceedings of the 2020 11th International Conference on Prognostics and System Health Management (PHM-2020 Jinan); IEEE, 2020; pp. 450–455.
  4. Zou, Y.; Yu, L.; Tso, G.K.F.; He, K. Risk forecasting in the crude oil market: A multiscale Convolutional Neural Network approach. Phys. A Stat. Mech. its Appl. 2020, 541, 123360, doi:10.1016/j.physa.2019.123360.Systems.; 2020; pp. 809–817 ISBN 9789811526954.

(Reviewer 2) Question 11:

  • Lines 27: What do “sample number” and MAPE mode” mean?
  • Lines 27-30: The description of the results achieved should be rewritten. It is not clear what the authors mean.
  • Line 54: The expression “length of the forecast time” is not clear, “forecasting horizon” would be better.
  • Line 72: The meaning of CNN should be explained.
  • Line 75: The meaning of AS-GCLSSVM should be explained.
  • Line 77: The meaning of GWO and CV should be explained.
  • Lines 85.86: “For some nonlinear systems, the fuzzy neural 85 network also needs to be combined with Fourier and other regression models” Some references should be provided to support this statement.
  • Line 77: The meaning of BPNN should be explained.
  • Line 93: The meaning of ELM-GA should be explained.
  • Line 104: The meaning of FA-KELM should be explained.
  • Line 115: The meaning of 2D-CNN should be explained.
  • Line 127: The meaning of TCN should be explained.
  • Line 130: The meaning of GRU should be explained.
  • Line 181: the concept of residual linking should be further explained.
  • Lines 227-229: How have the structure of kernel, filters and output data been defined? The authors should justify why they have used those values. They should also clarity what a filter is in this structure.
  • Lines 238-239: What do the authors mean with “…the same time of the week…”?
  • Line 284: Which is the difference between “...data for five weeks...” and “...electricity load data...“
  • Line 285: It should be Figure 9 instead of Figure 3.
  • Line 313: It should be Figure 10 instead of Figure 4.
  • (Author) Respond:

(4)(5)(6)(8)(9)(10)(11)(12)(13) We have added a list of abbreviations with their expanded forms (see page 16, line 470).

(1) We have divided the data into a set of samples. The samples consist of model inputs and model outputs. The inputs are the previous-week hourly load data, and the outputs are the forecast-day hourly load data. There are 1057 samples.

(2) We have rewritten the description of the obtained results. Please see page 1, line 31 of the revised manuscript.

(3) We have replaced “length of the forecast time” with “forecasting horizon.”

(7) We have added a reference to support this statement.

(14) “Residual linking” means shortcut, as shown in Figure 4.

(15) The structure of the output data depends on the forecasting horizon. The structure of the kernel and filters is not interpretable and is artificially set through experiments. The filter is a matrix, often used to extract the characteristics of the data. Refer to Figure 1 and (6) for the specific convolution process.

(16) “…the same time of the week…” means the same time on different days of the week.

(17) “...data for five weeks...” means “the hourly lag of 5 weeks ACF.” It is different from “...electricity load data...” On page 10, line 322 — “The hourly lag of 5 weeks and the daily lag of 2 weeks selected for the same time, as evaluated using the ACF, are shown in Figure 9.”

(18)(19) We have changed the numbering of the figures.

Round 2

Reviewer 2 Report

A final revision of language should be carried out in order to remove some minor errors.

A final revision of the paper structure should also be carried out as some figures and their captions are in different pages (figures 2, 11 and 15). On the other hand figures 14-17 should be included in the main text (for example before chapter 4. Conclusions), not after Apendix A.

Two Figure 14 appear in the text. Figures after the first "Figure 14" should be renumbered.

Author Response

Comments and Suggestions for Authors

A final revision of the language should be carried out to remove some minor errors.

A final revision of the paper structure should also be carried out as some figures, and their captions are on different pages (figures 2, 11, and 15). On the other hand figures, 14-17 should be included in the main text (for example, before chapter 4. Conclusions), not after Appendix A.

Two Figure 14 appear in the text. Figures after the first "Figure 14" should be renumbered.

  • (Reviewer 2) Question 1: A final revision of language should be carried out to remove some minor errors.

(Author) Respond:

Thank you for your comment. We have carefully checked and proofread the entire article's English grammar, and we invited a professor to check our manuscript; please see the revised manuscript.

  • (Reviewer 2) Question 2: A final revision of the paper structure should also be carried out as some figures and captions are on different pages (figures 2, 11, and 15).

(Author) Respond:

In response to this problem, we have readjusted some paragraphs and figures' positions to see these figures and their titles on the same page as possible. Please see the revised manuscript.

  • On the other hand figures, 14-17 should be included in the main text (for example, before chapter 4. Conclusions), not after Appendix A.

(Author) Respond:

Thank you for your comment. We have modified these figures' location and renumbered the order, and please see the revised manuscript.

  • (Reviewer 2) Question 3: Two Figure 14 appear in the text. Figures after the first "Figure 14" should be renumbered.

(Author) Respond:

We have modified these figures' location and renumbered the order, and please see the revised manuscript. Thanks for your valuable suggestions. I hope you have a nice day.